

# Emission Factors of Black Carbon and Co-pollutants from Diesel Vehicles in Mexico City

Miguel Zavala[1], Luisa T. Molina[1], Tara I. Yacovitch[2], Edward C. Fortner[2], Joseph R. Roscioli[2], Cody Floerchinger[2], Scott C. Herndon[2], Charles E. Kolb[2], Walter B. Knighton[3], Victor Hugo Paramo[4], Sergio Zirath[4], José Antonio Mejía[5], Aron Jazcilevich[6]

[1]Molina Center for Energy and the Environment, La Jolla, CA, 92037, USA
[2]Aerodyne Research, Inc., Billerica, MA, 01821, USA
[3]Department of Chemistry and Biochemistry, Montana State University, MT, 59717, USA.
[4]Instituto Nacional de Ecología y Cambio Climático, Mexico City, 04530, Mexico
[5]Environmental & Transport Consultant, Mexico City, Mexico
[6]Centro de Ciencias de la Atmosfera, Universidad Nacional Autónoma de México, Mexico City, 04510, Mexico

*Correspondence to*: Luisa Molina (ltmolina@mce2.org; ltmolina@mit.edu)

**Abstract.** Diesel-powered vehicles are intensively used in urban areas for transporting goods and people but can substantially contribute to high emissions of black carbon (BC), organic carbon (OC), and other gaseous pollutants. Strategies aimed at controlling mobile emissions sources thus have the potential to improve air quality as well as help mitigate impacts of air pollutants on climate, ecosystems, and human health. However, in developing countries there are limited data on the BC and OC emission characteristics of diesel-powered vehicles and thus there are large uncertainties in the estimation of the emission contributions from these sources. We measured BC, OC and other inorganic components of fine particulate matter (PM), as well as carbon monoxide (CO), nitrogen oxides ($NO_x$), sulfur dioxide ($SO_2$), ethane, acetylene, benzene, toluene, and C2-benzenes under real-world driving conditions of 20 diesel-powered vehicles encompassing multiple emission level technologies in Mexico City with the chasing technique using the Aerodyne mobile laboratory. Average BC emission factors ranged from 0.41-2.48 g/kg-fuel depending on vehicle type. The vehicles were also simultaneously measured using the cross-road remote sensing technique to obtain the emission factors of nitrogen oxide (NO), CO, total hydrocarbons, and fine PM, thus allowing the inter-comparison of the results from the two techniques. There is overall good agreement between the two techniques and both can identify high and low emitters but substantial differences were found in some of the vehicles, probably due to the ability of the chasing technique to capture a larger diversity of driving conditions in comparison to the remote sensing technique. A comparison of the results with the US-EPA MOVES-2014b model showed that the model underestimates CO, OC, and selected VOC species whereas there is better agreement for $NO_x$ and BC. Larger OC/BC ratios were found in comparison to ratios measured in California using the same technique, further demonstrating the need for using locally-obtained diesel-powered vehicle emission factors database in developing countries in order to reduce the uncertainty in the emissions estimates and to improve the evaluation of the effectiveness of emissions reduction measures.



## 1 Introduction

On-road mobile sources can substantially contribute to high emissions of black carbon (BC), organic carbon (OC), and other particulate matter (PM) components in urban areas. Although both gasoline and diesel powered vehicles are emitters of primary fine particulate matter, the available evidence indicates that when normalized to fuel consumption, PM emission factors are

more than an order of magnitude higher for heavy-duty diesel vehicles compared to light-duty gasoline vehicles (e.g., Ban-Weiss et al., 2008; Dallman et al., 2014). Freight tractor trailers, public transport buses, and heavy-duty trucks are typically powered by diesel fuel due to their high requirements of power, durability, and fuel efficiency. However, diesel-power vehicles can also contribute to high levels of nitrogen oxides ($NO_x$), carbon monoxide (CO), volatile organic compounds (VOCs) and other harmful co-pollutants. Thus, controlling diesel-powered mobile emissions has potential to improve air quality as well as

help mitigate impacts of air pollutants on climate, ecosystems, and human health.

Compared to gaseous pollutants emissions, direct measurements of emission factors for PM components from diesel-powered vehicles are less abundant. Until recently most of the measurements of PM from diesel-powered vehicles have been obtained using either semi-quantitative opacity-based techniques or by time-integrated gravimetric measurements that are subsequently

analyzed in the laboratory to estimate mass fractions of BC, OC, and other chemical PM components. In many of these studies, results are obtained using dynamometers to achieve pre-established engine-loads, standardized driving cycles, and controlled sampling conditions (e.g., Zhen et al., 2009; Cadle et al., 2009; Khalek et al., 2015). Recent technological advancements have allowed the direct measurement of BC emissions from diesel-powered vehicles under real-world driving conditions using mobile laboratories (e.g., Thornhill et al., 2010; Wang et al., 2012; Lau et al., 2015; Jezek et al., 2015), and tunnel studies (e.g.,

Geller et al., 2005; Ban-Weiss et al., 2008, 2009; Brimblecombe et al., 2015). Cross-road remote sensing studies and measurements obtained with on-board portable emission measurement systems (PEMS) have also been used to characterize $NO_x$, CO, hydrocarbons (HC), and other gaseous emissions from heavy-duty diesel vehicles (e.g., Burgard et al., 2006; Frey at al., 2008; He et al., 2010; Carslaw and Rhys-Thyler 2013).

Exhaust emissions measurements obtained using on-road or road-side mobile laboratories, traffic tunnel sampling, cross-road remote sensing, and PEMS sampling techniques vary substantially in their sampling size, sampling time, captured driving modes, and pollutants sampled. For example, traffic tunnel sampling and cross-road remote sensing studies can sample hundreds of vehicles in relatively short periods but are limited in the range of driving conditions captured. In contrast, on-road exhaust plume interception studies with mobile laboratories and PEMS can provide large amounts of information on emissions

under diverse driving conditions but are often limited in their sample size. Nevertheless, the overall results from the available studies have shown that there are important differences in the emission factors obtained under real-world driving conditions when compared to dynamometer-based studies. Furthermore, recent research suggests that in-use emissions of $NO_x$ are routinely underestimated relative to certification standards (Anenberg et al., 2017; Franco et al., 2014). The differences arise



because in real-world driving conditions there are multiple parameters (e.g., driving behavior, fuel quality, engine mechanical conditions, road conditions, etc.) that simultaneously affect the emission characteristics of on-road vehicles. These effects may not be properly captured under controlled tests (Ropkins et al., 2009). There have been some efforts to incorporate emissions and activity data obtained with PEMS into dynamometer-based tests to improve the representation of real-world driving

conditions for heavy-duty diesel trucks, but there are still substantial challenges for standardizing the certification and compliance testing procedures (e.g., Zhen et al., 2009; Giechaskiel et al., 2016; Maricq et al., 2016). As mobile emission inventories should aim to accurately represent real-world driving conditions, there is a continuing need to better characterize on-road emission factors using real-world sampling techniques.

Current estimates suggest that on-road diesel vehicles are a major source of BC and other submicron carbonaceous particles in many parts of the world (Bond et al., 2013). However, the estimates are highly uncertain due to different assumptions about emission factors and the fraction of high-emitting vehicles in developing countries' fleets. In Mexico, the most recent BC emissions estimates from the 2013 greenhouse gases and black carbon emission inventory (2013 GHG-BC MNEI), suggest that on-road vehicles contribute about 25% of the total 125 Gg annual BC emissions (SEMARNAT, 2015). However, due to

lack of locally obtained data, Mexico's BC and co-pollutants estimates for the diesel vehicle fleet were obtained using the default databases in the MOVES2014 EPA model (EPA, 2015) without adjusting emission factors or ancillary data. Therefore, there is a strong need to better characterize fine PM and gaseous pollutants emitted from diesel-powered vehicles in Mexico. In particular, the development of accurate emission factors and activity data for on-road vehicles is a critical step towards reducing uncertainties in Mexico's on-road emissions inventories.

In this pilot study we measured the fuel-based emission factors for BC, OC, CO, NO$_x$, and selected VOCs under real-world driving conditions for 20 on-road diesel vehicles in Mexico using the Aerodyne Research Inc. mobile laboratory (AML). The emission factors of NO, CO, HC, and fine PM were simultaneously measured using the cross-road remote sensing technique, thus allowing the inter-comparison of the results obtained by the two techniques. The sampled vehicles included service trucks,

metrobuses, turibuses, and inter-city urban buses encompassing EPA98, EPA03, EPA04, EURO3-5 emission level technologies. The results of this pilot study are useful to better understand the emission characteristics of the diesel vehicle fleet and to evaluate the emission factors used for the development of emissions inventories in Mexico, as well as to provide insights of diesel vehicle fleet emissions in other developing countries.

## 2 Methodology

Measurements were performed at Modulo 23, a large facility that is part of the Mexico City public transportation service (Red de Transporte de Pasajeros, or RTP), in collaboration with Mexico City Secretariat of Environment (Secretaría del Medio Ambiente, or SEDEMA) during February 25-28 of 2013, as part of the field measurement campaign to characterize the



emissions from key sources of Short-Lived Climate Forcers (SLCF-2013 Mexico). The Modulo 23 is typically used by RTP as a parking and maintenance facility for their public transport buses (see Fig. S1 in the Supplemental Material document). For this pilot project SEDEMA authorities redirected all of their scheduled RTP buses during the measurement period so that the parking area was empty and free of buses, except those selected for this study.

## 2.1 Sampling techniques

### 2.1.1 ARI mobile laboratory

The measurements were obtained using the AML by targeting on-road vehicles in "chase" and stationary road-side "exhaust plume-sampling" techniques following the procedures described in Zavala et al. (2006). Tested vehicles were driven on

prescribed routes inside and outside the Modulo 23 parking facility while the AML was positioned behind target diesel vehicles for continuously sampling their exhaust emissions with fast time response on-board instrumentation. For these on-road chase measurements the AML's velocity and acceleration were also recorded continuously as the AML trailed target vehicles sampling their exhaust plumes for a variety of driving conditions. Emissions ratios are obtained by correlating the sampled exhaust plume gaseous and particle signals with above background $CO_2$ and CO, which contain nearly all of pre-combustion

fuel carbon. Respective amounts of exhaust plume and background pollutant concentrations are determined by comparing background levels measured just before and after each plume encounter with those inside the exhaust plumes, effectively correcting for background and providing an exhaust emission ratio that can be used to obtain fuel-based emission factors (Zavala et al., 2006).

In addition to the on-road chase technique, the AML also employed the stationary road-side exhaust plume technique consisting of positioning the mobile laboratory downwind of the sampled target vehicles' exhaust. For instance, in collaboration with SEDEMA authorities, the mobile laboratory was strategically parked in one of the city's main Bus Rapid Transit (Metrobus) passenger stations to measure the emission plumes of incoming and departing Metrobuses. Only low-speed de-accelerating and accelerating plumes were sampled at this venue. A total of 101 Metrobuses were sampled at the passenger station,

encompassing multiple model years, manufacturers and engine emissions Tiers.

The measurement of vehicle emissions with the mobile laboratory is possible due to the use of high-time resolution on-board instrumentation that is capable of capturing the highly transient conditions of the sampled plumes. BC and OC were measured using a soot particle aerosol mass spectrometer (SP-AMS) developed by ARI (Onasch et al., 2012). The application of the SP-

AMS for the characterization of real-world vehicle emissions has been described in detail by Dallman et al., (2014). The SP-AMS uses laser-induced incandescence of absorbing soot particles to vaporize both the coatings and black carbon cores of exhaust soot particles within the ionization region of the AMS, thus providing a high sensitivity measurement of the refractory



BC (rBC) mass and the particle's organic and inorganic coating materials (see Petzold et al., 2013). For simplicity, we have referred rBC to BC in this manuscript. The detection limit in mass spectrum mode of the SP-AMS for BC and OC were 30 and 60 $ng/m^3$, respectively, with a nominal time resolution of 1 s. In addition to BC and OC, the SP-AMS measures other inorganic PM components including nitrates, sulfates, ammonium, and chlorides corresponding to a particle size range of 35 nm – 1 μm. In this paper, we refer to PM emission factors obtained with the mobile laboratory as the sum of BC, OC and inorganic components simultaneously measured with the SP-AMS for each sampled vehicle.

Additional instruments were deployed in the AML to characterize the gaseous pollutants of the sampled vehicles. Quantum Cascade Tunable Infrared Laser Differential Absorption Spectrometers (QC-TILDAS) were used to measure CO, $SO_2$, ethane ($C_2H_6$), and acetylene ($C_2H_2$), (Dallman et al., 2013). A Proton Transfer Reaction Mass Spectrometry (PTR-MS) operated using $H_3O^+$ as the ionization reagent (Rogers et al., 2006) was run in multiple ion detection mode to measure selected VOCs. Species measured with the PTR-MS included acetaldehyde, benzene, toluene, and C2-benzenes (sum of $C_8H_{10}$ isomers: xylenes + ethylbenzene and benzaldehyde). Two Thermo Electron 42i chemiluminescent detectors modified for fast-response measurements of NO and $NO_y$ and a LiCor 6262 Non-Dispersive Infrared (NDIR) instrument for $CO_2$ and water vapor were also used. Calibrations of these instruments were checked using certified gas standards. Other instruments on-board the mobile laboratory included a global positioning system (GPS), a sonic anemometer, and a video camera. Further details on the ARI instruments typical detection limits are presented in Table S1 of the supplemental material document.

### 2.1.2 Remote sensing

A remote sensing (RS) unit model 4650 developed by Environmental Systems Products was deployed in a location near the start of the prescribed routes inside the Modulo 23 (see Fig. S1 in the supplemental material document). The RS unit included both low (0.15 m) and high (3.9 m) level sampling elevations for measuring exhaust emissions, which is an important advantage when characterizing emissions from diesel-powered vehicles that have elevated tailpipes. The grade at the Modulo 23 is 0º. The location of the RS close to the start of the prescribed route was selected on the basis of obtaining an accelerating plume of the tested vehicle. Speed bar detectors were used to obtain vehicle speed and acceleration at the moment of passing through the RS unit. A video camera was placed down the road from the RS unit to take pictures of license plates when triggered.

In the RS, a NDIR exhaust gas analyzer with an optical filter of a wavelength known to be uniquely absorbed by the molecule of interest is placed in front of each detector, determining its specificity. The light source is shone across the road and reflected back. Reduction in the signal caused by absorption of light by the molecules of interest reduces the detector's signal, and thus the number of molecules of the pollutant can be inferred. The RS instrument measures $CO_2$, CO, total HC (as propane equivalents) using infrared light, whereas ultraviolet spectrometers are used for NO and $NO_2$. PM levels are not directly

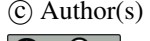



measured but inferred from a "smoke factor" estimated by the manufacturer from the ultraviolet and infrared absorption (Schuchmann et al., 2010). The target gas analyzers were calibrated daily with a mixture of certified gases. Technical specifications on the accuracy of the RS unit instruments are included in Table S2 of the supplemental material document.

## 2.2 Vehicles sampled

Vehicles sampled in this pilot study included 9 service trucks, 4 Metrobuses, 2 Turibuses, and 5 urban (RTP) buses, encompassing models years 1995 to 2011 and EPA98, EPA04, EURO3-5 technologies (Table 1 and Fig. S2). RTP urban buses are single 2 axle vehicles used for public transport with typical capacities of 60-90 passengers. RTP urban buses typically start operations very early in the morning and are continuously driven using designated intra-city routes until nighttime when they are returned to Modulo 23 for regular maintenance, refueling, and overnight parking. Thus, RTP urban buses are continuously

used throughout the day and often driven in low-speed but intense urban traffic conditions. Metrobuses are buses of one or two (merged) units that are used for transporting a large number of passengers (typical capacity is about 170 passengers) and have a dedicated (exclusive) driving lane on their route roads. The intra-city routes of Metrobuses are selected for connecting highly populated but largely separated areas in Mexico City using in-between passenger stations. Since no other vehicles are allowed to travel in the designated lanes, Metrobuses often are driven at higher speeds than the rest of the surrounding fleet

and are less affected by traffic.

Turibuses are double-decker buses that take passengers on guided tours through the main touristic landmarks of the city. Turibuses are usually driven at lower speeds than the Metrobuses, with gentler driving modes, and are well maintained. The service trucks tested were medium class 7 diesel trucks used for transporting goods for Coca Cola-FEMSA. The sampled

service trucks are typically driven in urban roads and are subject to intense traffic conditions. All vehicles sampled used ultra-low sulfur diesel (ULSD, < 15 ppm in S) except the Turibuses which used biodiesel with a B20 blend. All tested vehicles were ballasted in normal load operating conditions either with actual goods (for service trucks) or volunteers (for RTP buses, Metrobuses and Turibuses) during the measurements.

## 2.3 Data processing

Data from the three sampling techniques was processed to obtain fuel-based emission factors using established analytical protocols for the ARI mobile laboratory as described in detail in Zavala et al., (2006), and for the RS measurements as described in Bishop et al., (2008). In essence, the estimation of fuel-based emission factors EF for the two techniques is based on obtaining the mass ratio of the species of interest, $m_i$, to the total carbon mass found in above background $CO_2$ and $CO$, $m_{CO2}$ and $m_{CO}$, respectively, (and HC in the case of RS) normalized by the weight fraction carbon content of the diesel fuel $w_c$ (0.87)

as shown in Eq. (1):

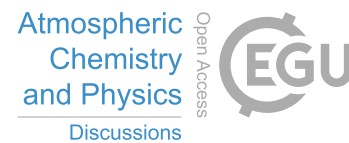

$$EF_i = \frac{\int m_i dt}{\int (m_{CO2} + m_{CO}) dt} w_c \tag{1}$$

For the AML technique the gaseous species mass ratio is obtained using the moles of the pollutant and the total moles of emitted carbon by multiplying with their respective molecular weights whereas the PM components measurements are directly obtained in $\mu gm^{-3}$, therefore the denominator units for the total carbon content need to be converted accordingly to $\mu gCm^{-3}$. For the RS technique, the ratio is obtained from the differences in number of molecules measured before and after the sampled vehicle. As described above, the PM levels in the RS technique are estimated from opacity measurements using a "smoke factor" to scale the absorption reading to grams of PM/grams of fuel and thus are semi-quantitative in nature but they are useful for inter-comparing vehicle emissions within the same experiment. For all the analysis, standard temperature and pressure conditions were used.

The important differences in the data analysis for the two techniques arise with the sampling frequency and thus with the integration periods ($\Delta t$) used to calculate the emission factors. In the mobile laboratory technique, an emission factor typically is obtained from multiple individual plume periods of 5-20 s depending on the truck velocity and wind conditions following the procedures described in Zavala et al., (2006). In this pilot study, each vehicle was sampled multiple times in prescribed routes with the mobile laboratory (see sampling size in Table 1) for about 3-10 minutes each time, therefore capturing hundreds of individual plumes measurements for each vehicle. In the RS technique, the light source travels multiple times back and forth between the detectors during the short time of the passing plume of the targeted truck and the integration period is close to 1 s. Thus, the resulting estimated emission factor represents a snapshot for the driving condition at the time when the vehicle is passing through the detector.

**3 Results**

Table 2 shows the average and 1-standard deviation of gaseous and PM fuel-based emission factors measured with the AML and the RS techniques for each of the sampled vehicles. The table also includes the results of the emission factors measured for the Metrobuses in stationary sampling mode as described above. Since the measurements were obtained in similar prescribed driving routes, the results show a wide range of average emission factors associated with each vehicle engine and emission control characteristics. The large standard deviations observed in Table 2 indicate that most vehicles presented high variability in their emission factors under the prescribed driving routes. The urban bus UB1 was visually identified as a high emitter during the experiments due to its intense black smoke exhaust plumes and this is confirmed by the much higher emission factors for this vehicle. Additionally, the relatively newer Dina urban bus with EURO5 technology had a malfunctioning selective catalytic reduction (SCR) device.





The average of the measured emission factors by vehicle type are shown in Fig. 1 and 2. Figure 1 shows a comparison of the CO and NO average emission factors obtained with the AML and the RS techniques whereas Fig. 2 shows the average emission factors of BC, OC, PM-inorganics, toluene, C2-benzenes, benzene, acetaldehyde, acetylene, and $SO_2$ measured with AML instruments. The inorganic component of PM was estimated as the sum of nitrate, sulfate, chloride and ammonium measured with the SP-AMS. For consistency in the comparisons, the emission factors shown for Metrobuses in Fig. 1 and 2 do not include data obtained in stationary sampling mode but only those obtained during the co-sampling of the two techniques.

The results show that the urban RTP buses produced the highest gaseous and PM emission factors. Conversely, the two sampled Turibuses running on biodiesel produced the lowest emission factors, particularly for BC, OC, and aromatics. The results also show that overall the emission factors measured with the remote sensing technique presented smaller variability with respect to those measured with the mobile laboratory. The higher variability observed with the mobile laboratory is likely the result of the larger range of driving conditions captured with this technique along the sampling routes, whereas the emission factors measured with the RS technique capture only the driving conditions at the time when the vehicle passes by the detectors.

Figure 3 shows a comparison of PM emission factors classified by vehicle control technology. Despite the small sampling size, the results indicate that there are marked differences between PM emissions depending on the vehicle's emissions control technology. Overall, the comparisons indicate lower PM emission factors due to improved control technology. The average PM emission factor decreases from 4.3 to 0.72 g/kg for vehicles with EURO3 to EURO5 technologies, respectively. However, the results also show that there is a large effect on the average PM emission factors when the data from the high polluting vehicle UB1 (EPA98) are included in the comparison. The average PM emission factor for the EPA98 category is reduced from 5.7 to 2.0 g/kg when the large emissions from this high-polluting vehicle are not included. Data from the single older 1995 ST7 vehicle with EPA94 technology was not included in Fig. 3 due to its relatively smaller sampling size (see Table 1). However, its average PM emission factor of 2.24 g/kg is slightly higher than the 2.0 g/kg average of the EPA98 technology excluding the high-emitting vehicle, consistent with the observed higher PM emission factors for vehicles with older technologies.

The sampling size in this pilot study is too small to be a representative sample of the entire Mexican fleet. Nevertheless, there are some vehicle age and type characteristics that make the results relevant. According to the 2013 GHG-BC MNEI the Mexican heavy-duty diesel fleet of about 810,000 vehicles is dominated by diesel trucks with gross vehicle weight (GVW) > 3 tons (~50.8%), followed by large trailer trucks (29.4%), urban buses (12.5%), and smaller trucks with GVW < 3 tons (4.5%) (Fig. S3 in supplemental material). Thus, the sampled service trucks, corresponding to diesel trucks with GVW>3 tons, the urban RTP buses and the Metrobuses belong to large categories of the diesel fleet. In addition, an analysis of the diesel-powered fleet distribution for Mexico City indicates that a large fraction of the in-use diesel vehicle fleet is relatively old and remains in-use for longer periods as compared to the gasoline vehicle fleet (Fig. S4 in supplement material). For example, about 61.5%



and 64.9% of the buses and diesel trucks with GVW>3 tons, respectively, are older than 10 years. The vehicle model years of the sampled service trucks (1995 – 2011) correspond to about 53.4% of the diesel trucks with GVW>3 tons fleet, whereas the model years of the sampled buses are relatively newer (2002-2011) and correspond to only about 36.6% of the buses fleet.

## 4 Discussions

### 4.1 Comparison between measurement techniques

As shown in Figure 1, the comparison of average CO and NO emission factors by vehicle type suggests an overall good agreement between the mobile laboratory and the remote sensing techniques, particularly for NO. However, rather than comparing the averages of emission factors, a proper comparison accounting for the actual co-sampling periods between the two techniques is required. Figure 4 shows the comparison of the individual CO and NO emission factors measured for each co-sampled vehicle. Since the remote sensing technique measures the emission factor of the sampled vehicle only while it passes through the detectors, only the emission factors obtained with the mobile laboratory ~10 seconds before and up to the corresponding actual moment of co-sampling with the remote sensing detector were considered for the comparison between the two techniques.

Figure 4 shows a linear but disperse correlation of the NO emission factors and a poor linear correlation of CO emission factors between the two techniques. Arguably, the results show an overall good agreement between the two techniques in terms of their ability to distinguish low and high CO and NO emitters; however, there is some indication that the agreement varies substantially by vehicle type. Whereas the overall coefficient of determination ($R^2$) between the two techniques is only 0.36 for CO, the coefficient increases to 0.92 and 0.85 for Metrobuses and service trucks, respectively. The lower CO emission factors for the UB1 high-emitter measured by the RS in comparison with the AML contributed significantly to the overall low correlation coefficient: not including the CO data for the UB1 in the comparison increases $R^2$ to 0.61. Similarly, the overall $R^2$ for NO emission factors between the two techniques is only 0.56 but it increases to 0.74 and 0.70 for service trucks and urban buses, respectively. Although the sampling size may be too small to provide a more precise quantification of the agreement between the two techniques, nevertheless, the results suggest that overall both techniques can be used to adequately distinguish between high and low emitters, but that distinction should consider the sampling efficiency by vehicle type.

### 4.2 Comparison with MOVES2014-Mexico model

MOVES2014 is currently the most advanced model for estimating on-road emissions in the US at national, state, county, and project level as it incorporates emissions data obtained from field studies over a wide range of vehicle types, pollutants, emission processes, fuel types, and operating modes (EPA 2015). A number of studies indicate that the use of the model can improve the emissions estimates of inventories in Mexico with respect to older emission models (Zavala et al., 2013; Guevara



et al., 2017). However, its efficient application to other countries requires the adjustment of multiple internal parameters, among which the emission factors databases are of key importance. A recent project was developed by the Eastern Research Group (ERG) to adjust the model's default emission factors and deterioration rates for the gasoline fleet using remote sensing data obtained in major Mexican cities (Koupal et al., 2016). The resulting model, MOVES2014-Mexico, also considers
Mexican vehicle emissions and fuel quality standards, vehicle population by age and state, fuel properties and fuel consumption. However, emission factors for the diesel fleet in the model were not adjusted using field measurements data.

The heavy-duty emission exhaust database for the MOVES2014 model's previous version (MOVES2010) was originally constructed using the results of several real-world in-use dedicated studies for gaseous pollutants, including: 1) measurements
of 124 trucks and buses with model years 1999 through 2007 using the Real-time On-road Vehicle Emissions Reporter (ROVER) PEMS system developed by EPA, and 2) measurements of 188 trucks with model years 1994 through 2003 using the Mobile Emissions Measurement System (MEMS) by West Virginia University (WVU). The current version of MOVES2014 builds upon these studies using two additional real-world studies: 1) the Heavy-Duty Diesel In-Use Testing (HDIU) program in which data was collected by manufacturers during normal operation for 243 diesel trucks of model years
2003-2009; and 2) the Houston Drayage Data (HDD) study in which the EPA collected emissions data from 27 trucks with model years 1991-2006 in drayage service using PEMS in the Houston-Galveston Area. Among other changes resulting from the emissions databases updates, MOVES2014 estimates higher $NO_x$ emission factors than MOVES2010 (EPA 2015).

Databases of PM emission factors in MOVES2014 were constructed from the CRC E-55/59 research program that consisted
in sampling 71 diesel vehicles with model years 1974-2004 (Clark et al., 2007). However, the PM speciation data was collected from only 9 different vehicles using the WVU's Transportable Heavy-Duty Vehicle Emissions Testing Laboratory (EPA 2014). Importantly, the measurements did not include transit buses and thus the PM emission factors for the urban bus vehicle category were proportionated using data from other measured vehicle types.

Figure 5 shows a comparison between emission factors measured with the AML and those from the MOVES2014-Mexico model in the "exhaust" emission process category. The figure compares the measured emission factors of urban buses and Metrobuses with those estimated for the Transit Bus vehicle category in the model. Measured emission factors of service trucks are compared against those estimated for the Single Unit Short-Haul Truck vehicle category in the model. Turibuses are not included in this comparison. In addition, in the comparison only the vehicle age groups in the MOVES2014-Mexico model
corresponding to those model years of the sampled vehicles are included.

The results indicate very good agreement between the modeled and measurement-based $NO_x$ emission factors for both buses and service trucks, but suggest a significant model underestimation of CO emission factors. Model-based BC emission factors are well within the observed values for service trucks but the results show higher variability in the measurements for the urban





buses and Metrobuses as compared to the model. The results also suggest a large underestimation of OC emission factors in the model for both buses and service trucks. Interestingly, despite the underestimation of OC there is a better agreement between the model and measurements for the total PM emission factor that results from a compensating effect of overestimation of the inorganic PM components in the model. The measured emission factors for acetaldehyde, benzene and

toluene were all much higher than those obtained from the MOVES2014-Mexico model, consistent with the observed underestimation of CO emission factors.

Overall, the model underestimated the CO, OC, and selected VOCs but had better agreement for $NO_x$ and BC emission factors. Due to the small sampling size in this pilot study, caution should be made when attempting to extrapolate the results from this

comparison to other vehicle categories and model years. Nevertheless, the results demonstrate the need for locally adjusting the emission factors database for the diesel vehicle fleet in the MOVES2014-Mexico model using real-world driving conditions to improve the emission estimates during inventory development.

### 4.3 Comparison with other studies

A recent study by Sheinbaum et al. (2015) investigated the impacts on PM and $NO_x$ emission levels when using B10 and B20 biodiesel blends for 3 EPA98 and 3 EPA04 urban RTP buses in Mexico. They found mixed results on the emission benefits depending on the technology and blend composition. The average reductions of PM for the three EPA04 buses were 66% and 36% using B10 and B20 blends, whereas the corresponding reductions for $NO_x$ where 4% and 8%. For the EPA98 buses PM increased 59% and 15% when using B10 and B20 blends, respectively, whereas $NO_x$ correspondingly increased 8% and 3%.

The two biodiesel TU1 and TU2 vehicles sampled in this study have EURO3 technology, similar to the ST4 and MT1 vehicles. Results in Table 2 show that the CO and $NO_x$ emissions factors of the TU1 and TU2 vehicles had no distinguishable differences with respect to the ST4 and MT1 vehicles. The major difference between the biodiesel fueled vehicles and other vehicles is observed in the much smaller BC and OC emission factors. Therefore, these results also suggest PM emission reduction benefits when using the biodiesel in vehicles with newer technology.

Table 3 compares the measured BC emission factors in this study with those reported in other parts of the world obtained with various sampling techniques. In 2006, the AML measured emissions from mobile sources in Mexico City using the chase technique and applied the positive matrix factorization (PMF) method to obtain an average fleet-wide emission factor of 1.4 g/kg for the diesel fleet (Thornhill et al., 2010). This value is well within the ranges of the measured emission factors in this

study and is similar to the values obtained in Beijing and Chongqing by Wang et al. (2012). Preble et al. (2015) found much smaller BC emission factors that corresponded to newer trucks with DPF and SCR control technologies. Our results also indicate that there is a strong effect of control technology on BC emissions. Nevertheless, the results also demonstrate that the information on the fraction of high emitters in the diesel fleet in developing countries is a key parameter for the construction





of emissions inventories. In addition, the values in Table 3 indicate that there is large variability of BC emission factors measured worldwide and at different times, highlighting the need for increasing the available datasets of emission factors obtained under real-world driving conditions to improve emissions inventory accuracy.

Dallman et al. (2014) obtained an average BC emission factor of $0.62 \pm 0.17$ in 2010 in San Francisco using also the SP-AMS instrument and found an OC/BC ratio of $0.31 \pm 0.1$ for the diesel fleet. The OC/BC emission ratios in this study are much higher: 0.59, 1.19, 1.26, and 1.56 for urban buses, Metrobuses, service trucks, and Turibuses, respectively. The biodiesel Turibuses presented the larger OC/BC ratio although their BC emission factors were the smallest of all sampled vehicles. The higher organic content of the emissions in the sampled Mexican vehicles with respect to those measured in California by
Dallman et al., (2014) illustrate the large emission differences in PM composition that can be found in diesel fleets around the world, thus further indicating the need for locally adjusting the emission factors databases in mobile emission models.

## 5 Conclusions

We present the results of the measurements of fuel-based emission factors for BC, OC, CO, $NO_x$, and selected VOCs for
diesel-powered service trucks, urban buses, Metrobuses and Turibuses in Mexico under real-world driving conditions using the AML and the remote sensing sampling techniques. The results showed higher PM emissions factors for urban buses with older technologies than for the other vehicle types and a marked dependency on vehicle emission control technology. These results further demonstrate the benefits of tighter Tier regulations and independent testing to verify the efficacy of reduced emissions standards for diesel vehicles.

The two biodiesel Turibuses presented smaller BC and OC emission factors. Although the effects from using biodiesel fuel could not be quantified in this study, the results suggest substantial emission benefits. Further dedicated studies with larger sampling size can help to quantify the benefits.

The comparison between the emission factors obtained with the two sampling techniques suggest that both techniques can be used to identify high and low vehicle emitters, but there are differences in sampling efficiency depending on the vehicle type sampled. In addition, higher variability was observed in the emission factors obtained with the mobile laboratory that likely results from the larger diversity of emission driving conditions captured with respect to the fixed-site remote sensing technique.

Comparison of the measured results with the emission factors estimated in the MOVES2014-Mexico model show that the model underestimates CO, OC, and selected VOC species but that there is better agreement for $NO_x$ and BC. The underestimation of organic components in the model is further supported by the larger OC/BC ratios found in comparison to



ratios measured elsewhere with the same sampling technique. These results further demonstrate the need to locally adjust the emission factors databases for the diesel vehicle fleet when the MOVES2014 model is applied in countries other than the US in order to reduce the uncertainty in the emissions estimates and to improve the evaluation of the effectiveness of emissions reduction measures.

Disclaimers:

The use of the MOVES2014-Mexico model in this paper is for illustrative purposes and should not be considered as an evaluation of the model's performance.

10 The authors declare that they have no conflict of interest.

**Acknowledgments**

The SLCF-2013 Mexico field measurement campaign was coordinated by the Molina Center for Energy and the Environment under UNEP Contract GFL-4C58. MZ and LTM acknowledge additional support from NSF Award 1560494. The authors would like to thank Coca Cola FEMSA, Red de Transporte de Pasajeros, Metrobus, and ADO Turibus for providing the sampled vehicles. Special thanks to SEDEMA for their strong logistical support during the measurements. The Project Team also would like to thanks Francisco Guardado from the Instituto Nacional de Ecología y Cambio Climático (INECC) for logistical support.

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



Table 1. Summary of sampled vehicle characteristics.

| Vehicle type | Vehicle ID | Make | Model year | Tier | Sampling size[1] | |
| --- | --- | --- | --- | --- | --- | --- |
| | | | | | AML | RS |
| Service truck | ST1 | Freightliner | 1998 | EPA98 | 22 | 3 |
| | ST2 | Freightliner | 1998 | EPA98 | 21 | 7 |
| | ST3 | International | 2011 | EPA04 | 14 | 6 |
| | ST4 | Freightliner | 2006 | EURO3 | 15 | 4 |
| | ST5 | HINO | 2011 | EURO4 | 14 | 4 |
| | ST6 | Kenworth | 2010 | EPA04 | 15 | 4 |
| | ST7 | Mercedes-Benz | 1995 | EPA94 | 4 | 4 |
| | ST8 | Freightliner | 1999 | EPA98 | 9 | 3 |
| | ST9 | Freightliner | 1999 | EPA98 | 8 | 3 |
| Urban bus | UB1 | International | 2002 | EPA98 | 38 | 8 |
| | UB2 | International | 2009 | EPA04 | 21 | 6 |
| | UB3 | Mercedes- Benz | 2002 | EPA98 | 29 | 6 |
| | UB4 | Mercedes- Benz | 2009 | EPA04 | 9 | 6 |
| | UB5 | Dina | 2013 | EURO5 | 15 | 15 |
| Metrobus | MT1 | Scania | 2005 | EURO3 | 9 | 5 |
| | MT2 | Volvo | 2009 | EURO4 | 9 | 8 |
| | MT3 | Mercedes- Benz | 2011 | EURO5 | 10 | 13 |
| | MT4 | Volvo | 2012 | EURO5 | 6 | 8 |
| Turibus | TU1 | Scania | 2006 | EURO3 | 6 | 5 |
| | TU2 | Scania | 2002 | EURO3 | 8 | 5 |

[1]Sampling size refers to the number of emission factors obtained with the mobile laboratory (AML) and remote

5  sensing (RS) unit from the prescribed routes. See text for explanation of the sampling periods for each technique.




Table 2. Measured on-road average fuel-based emission factors (g/kg fuel) measured.[1]

| ID | CO | | NO | | NOx[6] | HC[5] | SO2 | C2H2 | C2H4O | Benzene | Toluene | C2-B[2] | BC | OC | Inorg[2] | PM[3] |
|---|---|---|---|---|---|---|---|---|---|---|---|---|---|---|---|---|
| | AML | RS | AML | RS | AML | RS | AML | AML | AML | AML | AML | AML | AML | AML | AML | RS |
| ST1 | 24.3 (15) | 17.1 (1) | 13.5 (3) | 12.5 (1) | 23.4 (5) | 0.8 (0.2) | 0.16 (0.1) | 0.12 (0.1) | 0.15 (0.1) | 0.16 (0.2) | 0.24 (0.2) | 0.23 (0.2) | 0.59 (0.5) | 0.88 (0.8) | 0.14 (0.1) | 2.45 (0.6) |
| ST2 | 20.3 (10) | 15.2 (5) | 11.4 (3) | 9.8 (1) | 19.7 (3) | 2.0 (1.1) | 0.16 (0.1) | 0.06 (0.03) | 0.17 (0.1) | 0.10 (0.04) | 0.21 (0.1) | 0.19 (0.1) | 2.01 (1.4) | 1.49 (0.7) | 0.04 (0.01) | 3.17 (1.1) |
| ST3 | 17.0 (9) | 11.1 (1) | 24.9 (20) | 14.2 (1) | 36.1 (8) | 2.5 (0.9) | 0.40 (0.2) | 0.05 (0.04) | 0.24 (0.1) | 0.12 (0.1) | 0.23 (0.2) | 0.31 (0.4) | 0.71 (0.6) | 1.05 (1.3) | 0.09 (0.05) | 1.3 (0.2) |
| ST4 | 32.8 (18) | 23.4 (4) | 33.8 (25) | 35.8 (5) | 45.5 (14) | 8.0 (1.3) | 0.72 (0.4) | 0.12 (0.1) | 0.49 (0.7) | 0.26 (0.3) | 0.58 (0.8) | 0.52 (0.8) | 0.96 (0.5) | 1.11 (0.6) | 0.11 (0.03) | 2.06 (0.6) |
| ST5 | 61.9 (17) | 78.8 (7) | 11.5 (3) | 13.8 (3) | 17.6 (3) | 2.5 (0.6) | 0.63 (0.1) | 0.18 (0.06) | 0.54 (0.1) | 0.18 (0.1) | 0.30 (0.2) | 0.25 (0.1) | 0.72 (0.6) | 0.85 (0.4) | 0.06 (0.01) | 3.04 (0.4) |
| ST6 | 22.0 (24) | 15.4 (5) | 16.1 (3) | 18.5 (7) | 29.2 (6) | 2.5 (0.6) | 0.43 (0.1) | 0.08 (0.1) | 0.37 (0.3) | 0.18 (0.1) | 0.38 (0.3) | 0.30 (0.2) | 0.74 (0.7) | 0.92 (0.6) | 0.06 (0.01) | 2.11 (0.6) |
| ST7 | 23.1 (3) | 23.6 (7) | 18.4 (4) | 18.0 (2) | 28.9 (6) | 21.4 (3.1) | 0.35 (0.1) | 0.11 (0.03) | 0.50 (0.2) | 0.13 (0.04) | 0.18 (0.1) | 0.20 (0.02) | 0.14 (0.1) | 2.07 (0.2) | 0.03 (0.01) | 4.15 (0.8) |
| ST8 | 26.7 (4) | 26.2 (3) | 11.8 (2) | 13.1 (1) | 21.9 (6) | 2.4 (1.7) | 0.42 (0.2) | 0.11 (0.03) | 0.26 (0.1) | 0.12 (0.03) | 0.22 (0.1) | 0.18 (0.1) | 0.40 (0.2) | 0.80 (0.2) | 0.03 (0.01) | 2.28 (0.4) |
| ST9 | 33.8 (13) | 14.3 (5) | 14.9 (8) | 14.8 (5) | 21.0 (6) | 3.5 (0.6) | 0.51 (0.2) | 0.12 (0.1) | 0.38 (0.3) | 0.21 (0.2) | 0.65 (0.9) | 0.39 (0.4) | 2.19 (1.6) | 1.50 (0.8) | 0.08 (0.01) | 2.3 (0.4) |
| UB1 | 140.3 (131) | 34.3 (15) | 11.9 (5) | 10.1 (2) | 21.4 (7) | 5.2 (3.9) | 0.47 (0.4) | 0.39 (0.3) | 0.34 (0.2) | 0.30 (0.2) | 0.60 (0.7) | 0.58 (0.7) | 10.37 (11.6) | 4.50 (2.8) | 0.10 (0.03) | 11.45 (2.6) |
| UB2 | 23.5 (13) | 17.4 (14) | 20.3 (8) | 21.1 (4) | 39.9 (8) | 5.4 (6.5) | 0.21 (0.1) | 0.08 (0.04) | 0.23 (0.1) | 0.13 (0.1) | 0.23 (0.1) | 0.21 (0.1) | 0.30 (0.9) | 0.55 (0.7) | 0.04 (0.01) | 2.32 (1.8) |
| UB3 | 33.6 (15) | 17.2 (4) | 18.0 (7) | 17.1 (4) | 28.8 (10) | 4.4 (2.8) | 0.24 (0.4) | 0.08 (0.03) | 0.36 (0.4) | 0.14 (0.1) | 0.23 (0.2) | 0.28 (0.3) | 1.01 (1.4) | 0.85 (0.6) | 0.05 (0.02) | 1.77 (0.7) |
| UB4 | 23.4 (10) | 19.4 (9) | 18.2 (5) | 21.9 (6) | 31.2 (8) | 0.5 (0.1) | 0.12 (0.1) | 0.05 (0.03) | 0.15 (0.04) | 0.10 (0.03) | 0.13 (0.1) | 0.13 (0.1) | 0.58 (0.2) | 0.88 (0.3) | 0.04 (0.01) | 2.44 (1.1) |
| UB5 | 28.6 (14) | 15.3 (6) | 32.8 (7) | 35.8 (6) | 58.9 (7) | 6.0 (4.3) | 0.41 (0.1) | 0.05 (0.02) | 0.17 (0.1) | 0.11 (0.1) | 0.27 (0.2) | 0.17 (0.1) | 0.12 (0.1) | 0.57 (0.4) | 0.06 (0.02) | 1.17 (0.7) |
| MT1 | 35.6 (14) | 30.0 (11) | 15.4 (2) | 17.6 (3) | 29.0 (2) | 10.7 (6.2) | 0.16 (0.1) | 0.09 (0.03) | 0.18 (0.04) | 0.13 (0.03) | 0.19 (0.1) | 0.18 (0.1) | 3.64 (1.7) | 4.15 (1.8) | 0.03 (0.01) | 4.08 (1.2) |
| MT2 | 67.5 (57) | 32.9 (40) | 18.3 (4) | 24.6 (10) | 31.8 (5) | 2.4 (1.3) | 0.25 (0.1) | 0.08 (0.03) | 0.26 (0.1) | 0.17 (0.1) | 0.37 (0.2) | 0.33 (0.1) | 0.82 (0.4) | 0.91 (0.3) | 0.06 (0.01) | 0.73 (0.5) |
| MT3 | 43.6 (23) | 24.9 (16) | 21.1 (4) | 21.5 (5) | 37.0 (6) | 0.3 (0.1) | 0.22 (0.1) | 0.17 (0.1) | 0.26 (0.1) | 0.14 (0.05) | 0.90 (0.3) | 0.29 (0.1) | 0.27 (0.2) | 0.44 (0.1) | 0.03 (0.01) | 1.75 (0.9) |
| MT4 | 7.8 (5) | 28.4 (25) | 14.3 (3) | 12.7 (4) | 22.6 (4) | 6.0 (7.3) | 0.13 (0.03) | 0.06 (0.02) | 0.16 (0.04) | 0.07 (0.02) | 0.15 (0.04) | 0.12 (0.04) | 0.23 (0.1) | 0.40 (0.1) | 0.03 (0.01) | 2.21 (1.6) |
| MTs[4] | 53.9 (44) | | 13.5 (5) | | 21.2 (8) | | 0.18 (0.2) | 0.20 (0.2) | 0.51 (0.7) | 0.27 (0.3) | 0.50 (0.5) | 0.40 (0.4) | 0.99 (1.2) | 0.84 (0.7) | 0.14 (0.04) | |
| TU1 | 24.1 (7) | 33.2 (17) | 19.0 (5) | 21.9 (3) | 29.1 (6) | 12.7 (4.0) | 0.06 (0.03) | 0.16 (0.1) | 0.27 (0.1) | 0.09 (0.01) | 0.16 (0.1) | 0.16 (0.1) | 0.07 (0.02) | 0.50 (0.05) | 0.04 (0.01) | 3.6 (1.5) |
| TU2 | 22.1 (11) | 36.1 (12) | 16.1 (4) | 15.5 (2) | 26.7 (6) | 3.0 (0.7) | 0.17 (0.08) | 0.06 (0.04) | 0.23 (0.1) | 0.08 (0.01) | 0.12 (0.04) | 0.11 (0.03) | 0.76 (0.8) | 0.78 (0.3) | 0.04 (0.01) | 5.03 (1.8) |

[1]Numbers in parenthesis represent standard deviations. AML and RS stand for mobile laboratory and remote sensing techniques. Vehicle identification codes can be found in Table 1.

[2] C2-B correspond to the sum of $C_8H_{10}$ isomers. "Inorg" represents the sum of ammonium, chloride, sulfates, and nitrate measured with the SP-AMS.

[3] PM from remote sensing are obtained from a "smoke factor" applied to absorption measurements.

[4] MTs represent the measurements obtained in stationary mode for hundredths of Metrobuses. See text for further details.

[5] Total HC emission factors expressed as propane equivalents.

[6] NOx emission factors from the AML are expressed as NO2-equivalents



Table 3. Comparison of measurements of BC emission factors from diesel-powered sources.

| Location and sampling year | Source type | Sampling technique | Mean and SD [g/kg-fuel] | Reference |
|---|---|---|---|---|
| Mexico City, 2013 | Turibus | Chasing | 0.41 ± 0.7 | This study |
| Mexico City, 2013 | Metrobus | Chasing | 1.24 ± 1.8 | This study |
| Mexico City, 2013 | Urban bus | Chasing | 2.48 ± 7.3 | This study |
| Mexico City, 2013 | Service trucks | Chasing | 0.94 ± 1.1 | This study |
| Los Angeles, 1997 | HDDV | Tunnel | 1.3 | Kirchstetter et al., 1999 |
| San Francisco, 2005 | MDDT,HDDV | Tunnel | 0.78 ± 0.09 | Geller et al., 2005 |
| San Francisco, 1997, 2006 | MDDT, HDDV | Tunnel | 0.92 ± 0.07 | Ban-Weiss et al., 2008 |
| Mexico City, 2006 | Diesel fleet | Chasing[c] | 1.4 (1.3-1.6)[a] | Thornhill et al., 2010 |
| San Francisco, 2006 | HDDV | Tunnel | 1.7 ± 2.3 | Ban-Weiss et al., 2009 |
| Wilmington, CA, 2007 | HDDT | Chasing[c] | 0.5 (0.07–0.1)[b] | Park et al., 2011 |
| Beijing, 2009 | HDDT | Chasing | 2.2 (0.4-1.7)[a] | Wang et al., 2012 |
| Chongqing, 2010 | HDDT | Chasing | 1.6 (0.7-1.6)[a] | Wang et al., 2012 |
| Beijing, 2010 | HDDT | Chasing | 1.1 (0.2-0.8)[a] | Wang et al., 2012 |
| San Francisco, 2010 | HDDT | Tunnel | 0.62 ± 0.17 | Dallman et al., 2014 |
| Los Angeles, 2011 | HDDV freeways | Chasing[c] | 1.33 ± 0.33 | Hudda et al., 2013 |
| Slovenia, 2011 | Buses | Chasing | 0.4 (0.24–0.65)[b] | Jezek et al., 2015 |
| Oakland, CA, 2011-2013 | HDDT | Tunnel | 1.15 ± 0.19 0.09 ± 0.04 | Preble et al., 2015[d] |
| Hong Kong, 2013-2014 | HDDV | Chasing | 2.2 ± 0.3 | Lau et al., 2015 |
| Hong Kong, 2014 | Diesel fleet | Tunnel | 1.28 ± 0.76 | Brimblecombe et al., 2015 |

[a] Represent average and 1st and 3rd quartiles of data.

[b] mean and 95% confidence interval. HDDT: Heavy-duty diesel vehicle; HDDT: Heavy-duty diesel truck; MDDT:
Medium-duty diesel truck.

[c] Includes chasing and fleet average values.

[d] sampling of individual plumes on an overpass. High BC value represents 2009 fleet with 2% DPF and 0 % SCR installed, whereas low BC value represents 2010-2013 trucks with full DPF and SCR systems installed.




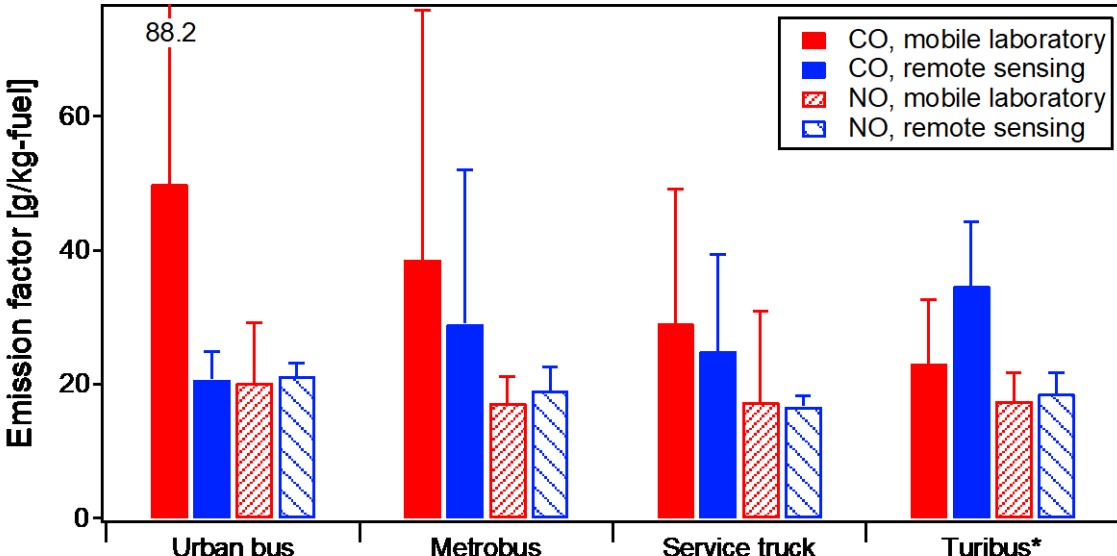

**Figure 1: Comparison of average fuel-based emission factors (g/kg fuel) between the mobile laboratory and remote sensing techniques by vehicle type. Variability bars represent 1 standard deviation of the observed values.\* Turibuses are fueled by biodiesel B20.**





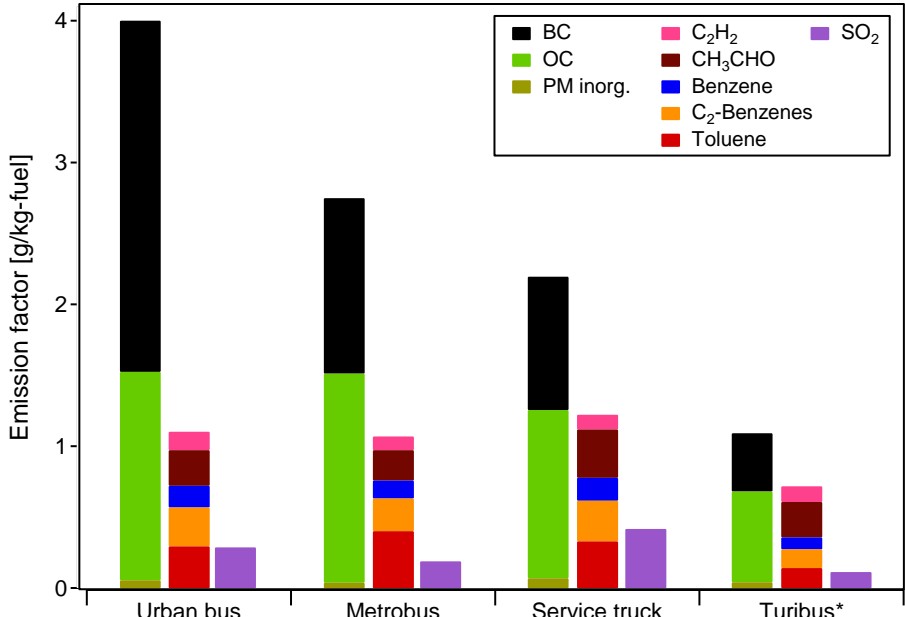

**Figure 2: Comparison of average VOCs, PM components, and SO2 fuel-based emission factors (g/kg fuel) measured with the mobile laboratory by vehicle type. PM Inorganics refers to the sum of ammonium, chloride, sulfates, and nitrate measured with the SP-AMS.\* Turibuses are fueled by biodiesel B20.**



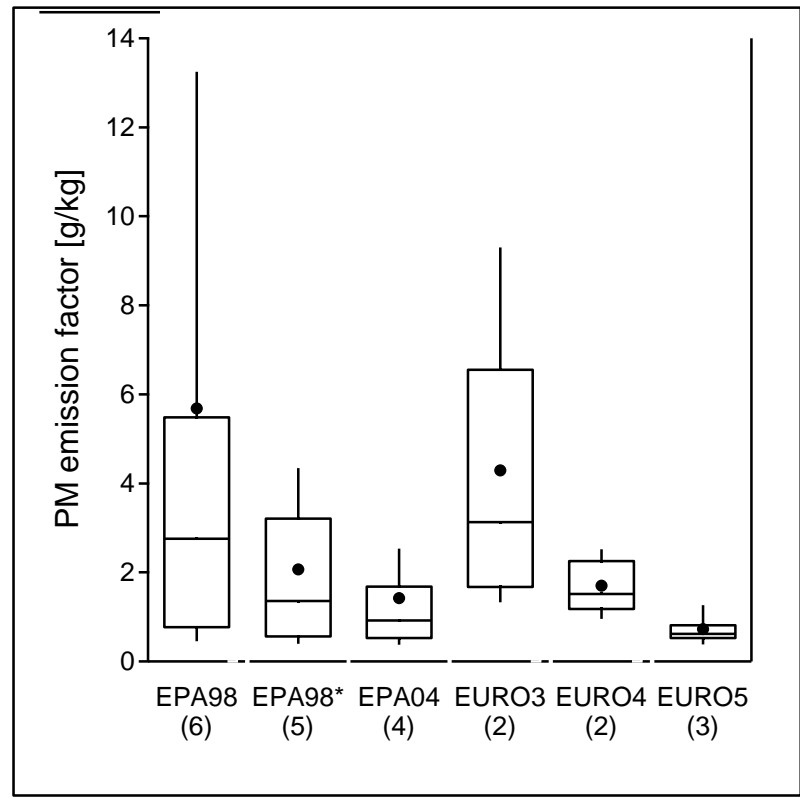

**Figure 3: Box plots of PM emission factors measured by control technology. The numbers in parenthesis represent the number of sampled vehicles. Upper vertical central lines, upper level box lines, middle horizontal lines, lower box lines, and lower vertical central lines represent 90%, 75%, 50%, 25%, and 10% of the data. The first box plot of the EPA98 technology category includes the high emitter vehicle UB1 (see Table 2) whereas the adjacent box plot does not include this vehicle. PM was obtained as the sum of BC, OC, chlorides, ammonium, sulfates, and nitrates components measured by the SP-AMS.**





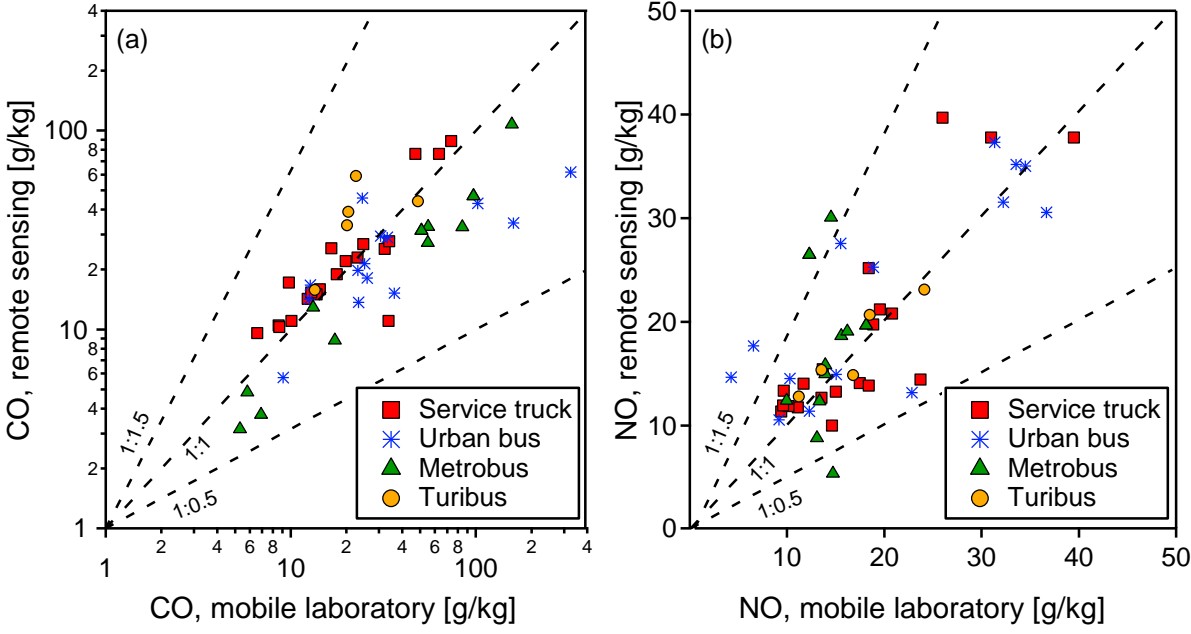

**Figure 4: Comparison of (a) CO and (b) NO fuel-based emission factors measured with the mobile laboratory and remote sensing techniques. Dashed lines represent 1:1.5, 1:1, and 1:0.25 lines.**





**Figure 5: Comparison between AML measurements and MOVES2014-Mexico emission factors.**