# Peer review of "Emission Factors of Black Carbon and Co-pollutants from Diesel Vehicles in Mexico City"

_Atmospheric Chemistry and Physics, 2017_

## Referee Comment (RC1) · Anonymous Referee #1 · 13 Sep 2017

**General comments**

Description: This discussion paper describes emission factors of diesel-powered trucks and buses in Mexico City measured using both the Aerodyne mobile laboratory and on-road remote sensing. The targeted compounds include CO, NOx, SO2, selected VOCs, PM, black carbon (BC), and particulate organic carbon (OC). The two methods produced similar results. BC emission factors were consistent with those measured in other studies, while the OC/BC ratio was larger than found in California. Emission factors generally agreed with those used in the EPA MOVES-2014b model for NOx and BC and were higher for CO, OC, and selected VOCs.

Relevance: Heavy-duty diesel-powered vehicles are responsible for substantial amounts of BC and NOx emissions, yet there are limited on-road measurements of

emissions from these vehicles. This work adds to the database of such measurements and shows that the chasing method with a mobile lab and the on-road remote sensing method produce comparable results, so it is fair to synthesize results across these different types of studies.

Assessment: The work contributes useful information about emissions from diesel engines. The writing and figures are very clear and informative. The paper illustrates the strengths and weaknesses of each of the two methods for measuring emission factors. The paper could be strengthened through some reorganization of the Results and Discussion and addition of statistical tests.

**Specific comments**

1. p. 4, line 10: A little more information about the prescribed driving routes and operation of the vehicles would be useful. What was the range of speeds? Were the engines always warmed up beforehand?

2. p. 7, lines 24-26: "Since the measurements were obtained in similar prescribed driving routes, the results show a wide range of average emission factors associated with each vehicle engine and emission control characteristics." The wording and logic are not quite right here. I think the authors intend to emphasize that driving conditions were very similar for all vehicles, so differences must reflect variability between engines and control systems. But later, they assert that there is large variability even for the same vehicle.

3. p. 8, line 7: The comparison of emission factors among different vehicle types begs for statistical tests of differences. This is true for the presentation of differences by control technology, too.

4. p. 8, line 26: The paragraph about the limitations of the sample size should be moved to the Discussion section.

5. p. 9, line 5: The comparison between the two methods in Section 4.1 seems it

belongs more in the Results section than in the Discussion section because it is a straightforward presentation of results that address one of the objectives of the study.

6. p. 9, line 11: For comparison of the two methods, the authors chose to use the 10 seconds of AML data leading up the instant of remote sensing, which lasted 1 second. Why not isolate the 1-2 seconds of AML data that best correspond to when the remote sensing measurement was captured?

7. p. 11, line 15: I assume that all the vehicles tested in this study were running on petroleum diesel, so results for B10 and B20 biodiesel are irrelevant to the present study and do not merit mention here, or they require greater justification for inclusion in the comparison.

8. p. 12, line 10: Can the authors comment on why there are differences in the OC/BC ratio compared to that found in other studies? Might altitude explain some of it or dilution? The mobile lab and remote sensing detect fresher, less diluted plumes compared to tunnel studies.

9. Table 2: This could be moved to the supplemental information, as a more digestible summary of the results appears in the figures. Footnote 4 mentions "hundredths" of Metrobuses. Should this be 101 Metrobuses, the number sampled?

10. Figure 4: The NO figure shows small variability in mobile lab measurements and much larger variability in remote sensing measurements (large spread in the y-axis direction). This does not comport with Fig. 1, which shows similar variability in the NO emission factors measured by both the mobile lab and remote sensing. Is it because these data points are limited to a much shorter period?

**Technical corrections**

11. p. 4, line 10: The wording "the AML was positioned behind target diesel vehicles" makes it sound like the AML was stationary or attached to the vehicles. I suggest something like, "the AML followed behind target diesel vehicles," instead.

12. p. 5, lines 1-2: Rewrite "we have referred rBC to BC in this manuscript. . .."

13. p. 5, line 2: "detection limit" should be "detection limits".

14. p. 7, line 26: Change "observed" to "reported".

———————————————————

---

## Referee Comment (RC2) · Anonymous Referee #2 · 30 Sep 2017

This is a very good paper with valuable information about diesel engine emission factors that can be used to prepare Mexican emission inventories. The experimental techniques used have been proven before and the results published in scientific journals. The paper is well written and the presentation of results is good. There are several questions I suggest to clarify before the paper is considered for publication. I enclose the reviewed file with comments and questions for your consideration. Please let me know if there is any doubt about my comments.

Please also note the supplement to this comment:
https://www.atmos-chem-phys-discuss.net/acp-2017-695/acp-2017-695-RC2-supplement.pdf

[Figure]

**Supplement:**

[Figure]

[revised manuscript text omitted]
 AML | CO RS | NO AML | NO RS | $NO_x$[6] AML | HC[5] RS | $SO_2$ AML | $C_2H_2$ AML | $C_2H_4O$ AML | Benzene AML | Toluene AML | C2-B[2] AML | BC AML | OC AML | Inorg[2] AML | PM[3] RS |
|---|---|---|---|---|---|---|---|---|---|---|---|---|---|---|---|---|
| ST1 | 24.3 (15) | 17.1 (1) | 13.5 (3) | 12.5 (1) | 23.4 (5) | 0.8 (0.2) | 0.16 (0.1) | 0.12 (0.1) | 0.15 (0.1) | 0.16 (0.2) | 0.24 (0.2) | 0.23 (0.2) | 0.59 (0.5) | 0.88 (0.8) | 0.14 (0.1) | 2.45 (0.6) |
| ST2 | 20.3 (10) | 15.2 (5) | 11.4 (3) | 9.8 (1) | 19.7 (3) | 2.0 (1.1) | 0.16 (0.1) | 0.06 (0.03) | 0.17 (0.1) | 0.10 (0.04) | 0.21 (0.1) | 0.19 (0.1) | 2.01 (1.4) | 1.49 (0.7) | 0.04 (0.01) | 3.17 (1.1) |
| ST3 | 17.0 (9) | 11.1 (1) | 24.9 (20) | 14.2 (1) | 36.1 (8) | 2.5 (0.9) | 0.40 (0.2) | 0.05 (0.04) | 0.24 (0.1) | 0.12 (0.1) | 0.23 (0.2) | 0.31 (0.4) | 0.71 (0.6) | 1.05 (1.3) | 0.09 (0.05) | 1.3 (0.2) |
| ST4 | 32.8 (18) | 23.4 (4) | 33.8 (25) | 35.8 (5) | 45.5 (14) | 8.0 (1.3) | 0.72 (0.4) | 0.12 (0.1) | 0.49 (0.7) | 0.26 (0.3) | 0.58 (0.8) | 0.52 (0.8) | 0.96 (0.5) | 1.11 (0.6) | 0.11 (0.03) | 2.06 (0.6) |
| ST5 | 61.9 (17) | 78.8 (7) | 11.5 (3) | 13.8 (3) | 17.6 (3) | 2.5 (0.6) | 0.63 (0.1) | 0.18 (0.06) | 0.54 (0.1) | 0.18 (0.1) | 0.30 (0.2) | 0.25 (0.1) | 0.72 (0.6) | 0.85 (0.4) | 0.06 (0.01) | 3.04 (0.4) |
| ST6 | 22.0 (24) | 15.4 (5) | 16.1 (3) | 18.5 (7) | 29.2 (6) | 2.5 (0.6) | 0.43 (0.1) | 0.08 (0.1) | 0.37 (0.3) | 0.18 (0.1) | 0.38 (0.3) | 0.30 (0.2) | 0.74 (0.7) | 0.92 (0.6) | 0.06 (0.01) | 2.11 (0.6) |
| ST7 | 23.1 (3) | 23.6 (7) | 18.4 (4) | 18.0 (2) | 28.9 (6) | 21.4 (3.1) | 0.35 (0.1) | 0.11 (0.03) | 0.50 (0.2) | 0.13 (0.04) | 0.18 (0.1) | 0.20 (0.02) | 0.14 (0.1) | 2.07 (0.2) | 0.03 (0.01) | 4.15 (0.8) |
| ST8 | 26.7 (4) | 26.2 (3) | 11.8 (2) | 13.1 (1) | 21.9 (6) | 2.4 (1.7) | 0.42 (0.2) | 0.11 (0.03) | 0.26 (0.1) | 0.12 (0.03) | 0.22 (0.1) | 0.18 (0.1) | 0.40 (0.2) | 0.80 (0.2) | 0.03 (0.01) | 2.28 (0.4) |
| ST9 | 33.8 (13) | 14.3 (5) | 14.9 (8) | 14.8 (5) | 21.0 (6) | 3.5 (0.6) | 0.51 (0.2) | 0.12 (0.1) | 0.38 (0.3) | 0.21 (0.2) | 0.65 (0.9) | 0.39 (0.4) | 2.19 (1.6) | 1.50 (0.8) | 0.08 (0.01) | 2.3 (0.4) |
| UB1 | 140.3 (131) | 34.3 (15) | 11.9 (5) | 10.1 (2) | 21.4 (7) | 5.2 (3.9) | 0.47 (0.4) | 0.39 (0.3) | 0.34 (0.2) | 0.30 (0.2) | 0.60 (0.7) | 0.58 (0.7) | 10.37 (11.6) | 4.50 (2.8) | 0.10 (0.03) | 11.45 (2.6) |
| UB2 | 23.5 (13) | 17.4 (14) | 20.3 (8) | 21.1 (4) | 39.9 (8) | 5.4 (6.5) | 0.21 (0.1) | 0.08 (0.04) | 0.23 (0.1) | 0.13 (0.1) | 0.23 (0.1) | 0.21 (0.1) | 0.30 (0.9) | 0.55 (0.7) | 0.04 (0.01) | 2.32 (1.8) |
| UB3 | 33.6 (15) | 17.2 (4) | 18.0 (7) | 17.1 (4) | 28.8 (10) | 4.4 (2.8) | 0.24 (0.4) | 0.08 (0.03) | 0.36 (0.4) | 0.14 (0.1) | 0.23 (0.2) | 0.28 (0.3) | 1.01 (1.4) | 0.85 (0.6) | 0.05 (0.02) | 1.77 (0.7) |
| UB4 | 23.4 (10) | 19.4 (9) | 18.2 (5) | 21.9 (6) | 31.2 (8) | 0.5 (0.1) | 0.12 (0.1) | 0.05 (0.03) | 0.15 (0.04) | 0.10 (0.03) | 0.13 (0.1) | 0.13 (0.1) | 0.58 (0.2) | 0.88 (0.3) | 0.04 (0.01) | 2.44 (1.1) |
| UB5 | 28.6 (14) | 15.3 (6) | 32.8 (7) | 35.8 (6) | 58.9 (7) | 6.0 (4.3) | 0.41 (0.1) | 0.05 (0.02) | 0.17 (0.1) | 0.11 (0.1) | 0.27 (0.2) | 0.17 (0.1) | 0.12 (0.1) | 0.57 (0.4) | 0.06 (0.02) | 1.17 (0.7) |
| MT1 | 35.6 (14) | 30.0 (11) | 15.4 (2) | 17.6 (3) | 29.0 (2) | 10.7 (6.2) | 0.16 (0.1) | 0.09 (0.03) | 0.18 (0.04) | 0.13 (0.03) | 0.19 (0.1) | 0.18 (0.1) | 3.64 (1.7) | 4.15 (1.8) | 0.03 (0.01) | 4.08 (1.2) |
| MT2 | 67.5 (57) | 32.9 (40) | 18.3 (4) | 24.6 (10) | 31.8 (5) | 2.4 (1.3) | 0.25 (0.1) | 0.08 (0.03) | 0.26 (0.1) | 0.17 (0.1) | 0.37 (0.2) | 0.33 (0.1) | 0.82 (0.4) | 0.91 (0.3) | 0.06 (0.01) | 0.73 (0.5) |
| MT3 | 43.6 (23) | 24.9 (16) | 21.1 (4) | 21.5 (5) | 37.0 (6) | 0.3 (0.1) | 0.22 (0.1) | 0.17 (0.1) | 0.26 (0.1) | 0.14 (0.05) | 0.90 (0.3) | 0.29 (0.1) | 0.27 (0.2) | 0.44 (0.1) | 0.03 (0.01) | 1.75 (0.9) |
| MT4 | 7.8 (5) | 28.4 (25) | 14.3 (3) | 12.7 (4) | 22.6 (4) | 6.0 (7.3) | 0.13 (0.03) | 0.06 (0.02) | 0.16 (0.04) | 0.07 (0.02) | 0.15 (0.04) | 0.12 (0.04) | 0.23 (0.1) | 0.40 (0.1) | 0.03 (0.01) | 2.21 (1.6) |
| MTs[4] | 53.9 (44) |  | 13.5 (5) |  | 21.2 (8) |  | 0.18 (0.2) | 0.20 (0.2) | 0.51 (0.7) | 0.27 (0.3) | 0.50 (0.5) | 0.40 (0.4) | 0.99 (1.2) | 0.84 (0.7) | 0.14 (0.04) |  |
| TU1 | 24.1 (7) | 33.2 (17) | 19.0 (5) | 21.9 (3) | 29.1 (6) | 12.7 (4.0) | 0.06 (0.03) | 0.16 (0.1) | 0.27 (0.1) | 0.09 (0.01) | 0.16 (0.1) | 0.16 (0.1) | 0.07 (0.02) | 0.50 (0.05) | 0.04 (0.01) | 3.6 (1.5) |
| TU2 | 22.1 (11) | 36.1 (12) | 16.1 (4) | 15.5 (2) | 26.7 (6) | 3.0 (0.7) | 0.17 (0.08) | 0.06 (0.04) | 0.23 (0.1) | 0.08 (0.01) | 0.12 (0.04) | 0.11 (0.03) | 0.76 (0.8) | 0.78 (0.3) | 0.04 (0.01) | 5.03 (1.8) |

[revised manuscript text omitted]

---

## Author Comment (AC1) · 8 Nov 2017

General comments

Description: This discussion paper describes emission factors of diesel-powered trucks and buses in Mexico City measured using both the Aerodyne mobile laboratory and on-road remote sensing. The targeted compounds include CO, NOx, SO2, selected VOCs, PM, black carbon (BC), and particulate organic carbon (OC). The two methods produced similar results. BC emission factors were consistent with those measured in other studies, while the OC/BC ratio was larger than found in California. Emission factors generally agreed with those used in the EPA MOVES-2014b model for NOx and BC and were higher for CO, OC, and selected VOCs.

Relevance:

Heavy duty diesel-powered vehicles are responsible for substantial amounts of BC and NOx emissions, yet there are limited on-road measurements of emissions from these vehicles. This work adds to the database of such measurements and shows that the chasing method with a mobile lab and the on-road remote sensing method produce comparable results, so it is fair to synthesize results across these different types of studies.

Assessment: The work contributes useful information about emissions from diesel engines. The writing and figures are very clear and informative. The paper illustrates the strengths and weaknesses of each of the two methods for measuring emission factors. The paper could be strengthened through some reorganization of the Results and Discussion and addition of statistical tests.

We thank the reviewer for the constructive comments on the paper.

Specific comments

1. p. 4, line 10: A little more information about the prescribed driving routes and operation of the vehicles would be useful. What was the range of speeds? Were the engines always warmed up beforehand?

We thank the reviewer for this suggestion as this will allow the results to be more adequately compared to future studies of emission characteristics of diesel vehicles in Mexico. We have now included a more detailed description of the driving conditions (the range of speeds and accelerations of the vehicle sampled) during the tests in the supplemental material document.

2. p. 7, lines 24-26: "Since the measurements were obtained in similar prescribed driving routes, the results show a wide range of average emission factors associated with each vehicle engine and emission control characteristics." The wording and logic are not quite right here. I think the authors intend to emphasize that driving conditions were very similar for all vehicles, so differences must reflect variability

between engines and control systems. But later, they assert that there is large variability even for the same vehicle.

We thank the reviewer for calling our attention to this ill-constructed phrase. As pointed out, we want to emphasize that the driving conditions were very similar for all vehicles. We have modified the paragraph accordingly:

"Since the measurements were obtained under similar prescribed driving routes, differences in results mainly reflect variability among vehicle engines and emission control characteristics."

As stated, the results indicate that even after controlling for driving routes the observed variability of emission factors still can be large. This is in agreement with current understanding of real-world emissions as compared to laboratory-based studies and the growing acknowledgment that engine performance can produce large variability under real-world operation conditions.

3. p. 8, line 7: The comparison of emission factors among different vehicle types begs for statistical tests of differences.  This is true for the presentation of differences by control technology, too.

We thank the reviewer for this valuable suggestion. We have now performed statistical testing for the significance of the results between vehicle types, by control technology, and between measurement techniques.

We have analyzed the statistical significance between control technologies for $PM_{2.5}$ EF using non-paired non-parametric Wilcoxon Rank tests and found that with a 95% confidence level the results for the EPA98 and EPA04 are significantly different. Similarly, differences between EURO3, EURO4, and EURO5 EF were found to be significantly different with a 95% confidence level. However, the rest of the tests indicated that the results for the EPA98 and the EURO3 (the older technologies sampled) were not significantly different with a 95% confidence level. Therefore, the following paragraph has been included in the text:

"Non-paired Wilcoxon Rank tests indicate that there is statistically significant difference (at the 0.05 significance level) between the $PM_{2.5}$ emission factors obtained for the EPA98 and EPA04 control technologies as well as among the EURO3, EURO4, and EURO5 technologies. However, the results for the EPA98 and the EURO3 technologies were not significantly different."

We similarly performed non-paired Wilcoxon Rank tests for comparing the emission factors by vehicle type for each pollutant. We found that CO, $NO_x$, and $SO_2$ from service trucks, urban buses, and Metrobuses were significantly different among them, whereas their VOCs measured ($C_2H_2$, acetaldehyde, benzene, toluene, C2-benzenes), and PM components (BC, OC, and inorganics) were not statistically significantly different. On the contrary, VOCs and PM-components emission factors obtained from Turibuses were statistically different from the corresponding emission factors from service trucks, urban buses, and Metrobuses. Thus we have included the following paragraph:

"Non-paired Wilcoxon Rank test indicate that there is statistically significant difference (at the 0.05 significance level) between emission factors from service trucks, urban buses, and Metrobuses for the CO, $NO_x$, and $SO_2$ pollutants, whereas their corresponding VOCs, BC, OC, and PM-inorganic emission factors were not significantly different. VOCs, BC, and PM-inorganic emission factors from biodiesel-fueled Turibuses were significantly different from the corresponding emission factors from service trucks, urban buses, and Metrobuses."

In addition to the analysis suggested by the reviewer we also evaluated the statistical significance of the results for CO and NO emission factors that were obtained with both the chasing and the remote sensing techniques. Since these represent co-sampled data we used paired t-test with a 0.05 significance level. The results indicate that in both cases of CO and NO co-samplings there is no significant difference between the results obtained by the two measurement techniques, with a confidence level of 95%. Therefore, we have now added the following paragraphs to the results:

"Paired t-tests indicate that there is no statistical significant difference (at the 0.05 significance level) between the two measurement techniques for both cases of CO and NO emission factors."

4. p. 8, line 26: The paragraph about the limitations of the sample size should be moved to the Discussion section.

As suggested by the reviewer, we have moved the discussion on the limitations of the sample size to the Discussion section.

5. p. 9, line 5: The comparison between the two methods in Section 4.1 seems it belongs more in the Results section than in the Discussion section because it is a straightforward presentation of results that address one of the objectives of the study.

We have moved the comparison of the two methods to the Results section.

6. p. 9, line 11: For comparison of the two methods, the authors chose to use the 10 seconds of AML data leading up the instant of remote sensing, which lasted 1 second. Why not isolate the 1-2 seconds of AML data that best correspond to when the remote sensing measurement was captured?

For the estimation of the emission factors using the chasing technique, the second-by-second measurements are integrated over a time period to account for the dispersion of the emission plume. Thus, if too few data points are included in the integration the resulting emission factor may not properly reflect the plume development and unnecessary uncertainty is introduced in the analysis. Based on our past experience with data analysis of this technique, we consider a good conservative

number of data points for plume development is about 10 seconds as the basis for the choice of integration time. Thus, we have now complemented the following sentence:

"Since the remote sensing technique measures the emission factor of the sampled vehicle only while it passes through the detectors, only the emission factors obtained with the mobile laboratory ~10 seconds before and up to the corresponding actual moment of co-sampling with the remote sensing detector were considered for the comparison between the two techniques. Thus, we assume that a time period of 10 seconds is sufficient to capture a large portion of the emission plume sampled by the mobile laboratory."

7. p. 11, line 15: I assume that all the vehicles tested in this study were running on petroleum diesel, so results for B10 and B20 biodiesel are irrelevant to the present study and do not merit mention here, or they require greater justification for inclusion in the comparison.

In the discussion section of the paper we compare our results on the emission factors from biodiesel vehicles to the only other available literature study of similar measurements in Mexico that used B10 and B20 blends. We believe that, given the very limited information currently available, the comparison information and discussion presented is a valuable addition and thus we have decided to keep it in the manuscript.

8. p. 12, line 10: Can the authors comment on why there are differences in the OC/BC ratio compared to that found in other studies? Might altitude explain some of it or dilution? The mobile lab and remote sensing detect fresher, less diluted plumes compared to tunnel studies.

There are several possible reasons why the results show higher OC/BC ratios in comparison to other studies. These include differences in conditions derived from the environment (e.g., altitude, temperature), technical sampling methods (capturing fresh versus more diluted emissions), and diesel fuel composition. Unfortunately it is not possible from our results to quantitatively assess the contributions from these factors as it is beyond the scope of this study. Dedicated experiments controlling for these factors as well as vehicle technology and driving conditions could help to quantify the impacts of these factors. Nevertheless, it is possible to argue that the higher OC/BC content in the Mexican results obtained with the mobile laboratory are not due to differences in the sampling technique used in tunnel studies. As the reviewer pointed out, the former capture more fresh emissions than tunnel studies and, therefore, secondary formation of organic aerosols in the air masses would only increase the OC/BC during in the tunnel study sampling (Massoli et al., 2012), which is in the opposite direction needed to explain the observed differences.

No samples were obtained in our study of the diesel fuel employed, and thus it is not possible to know its exact organic chemical composition and its effects on emissions. Although a detailed chemical composition of diesel fuel by PEMEX (Mexican National Oil Company) is not publicly available, an official

report indicates a predominant fraction of paraffin compounds of linear molecular chains with 11 to 12 carbons and a maximum 30% (in volume) of aromatics (IMP, 2014). In principle, a dedicated experiment could be set up to investigate the effects of OC formation due to differences in PEMEX's diesel fuel composition, but this is beyond the scope of this study. We have therefore included the following paragraph:

"Several factors including driving conditions, vehicle technology, and diesel fuel composition can contribute to the observed differences, but the quantification of these contributions is beyond the scope of this study. Nevertheless, the higher organic content of the emissions in the sampled Mexican vehicles with respect to those measured in California by Dallman et al., (2014) illustrate the large emission differences in PM composition that can be found in diesel fleets around the world, thus further indicating the need for locally adjusting the emission factors databases in mobile emission models."

References:

IMP, Instituto Mexicano del Petroleo: Factores de Emision para los diferentes tipos de combustibles fosiles que se consumen en Mexico. Informe Tecnico F.61157.02.005. Dirección de Servicios de Ingeniería Gerencia de Servicios en Ingeniería Región Centro-Norte. 2014. Available: http://www.inecc.gob.mx/descargas/cclimatico/2014_inf_parc_tipos_comb_fosiles.pdf

Paola Massoli , Edward C. Fortner , Manjula R. Canagaratna , Leah R. Williams , Qi Zhang , Yele Sun , James J. Schwab , Achim Trimborn , Timothy B. Onasch , Kenneth L. Demerjian , Charles E. Kolb , Douglas R. Worsnop & John T. Jayne (2012) Pollution Gradients and Chemical Characterization of Particulate Matter from Vehicular Traffic near Major Roadways: Results from the 2009 Queens College Air Quality Study in NYC, Aerosol Science and Technology, 46:11, 1201-1218, DOI: 10.1080/02786826.2012.701784.

9. Table 2: This could be moved to the supplemental information, as a more digestible summary of the results appears in the figures. Footnote 4 mentions "hundredths" of Metrobuses. Should this be 101 Metrobuses, the number sampled?

We agree that the information shown in Table 2 is somewhat dense. However, we believe it is important to present directly in the main manuscript a summary table of the average results obtained with both techniques. Therefore, we have decided to maintain Table 2 in the main text of the manuscript.

We thank the reviewer for the suggested edit on the footnote 4, the change has been made.

10. Figure 4: The NO figure shows small variability in mobile lab measurements and much larger variability in remote sensing measurements (large spread in the y-axis direction). This does not comport

with Fig. 1, which shows similar variability in the NO emission factors measured by both the mobile lab and remote sensing. Is it because these data points are limited to a much shorter period?

As pointed out by the reviewer, the differences in variabilities shown in Figures 1 and 4 are the result of the chosen periods for the comparisons but also on the averaging of results. Figure 1 shows "smaller" variability than Figure 4 because, as described in the manuscript, the figure is based on the averages of emission factors obtained from each vehicle, whereas Figure 4 shows the comparison of individual emission factors whenever they were co-sampled by the two techniques for the same vehicle.

Technical corrections

11. p. 4, line 10: The wording "the AML was positioned behind target diesel vehicles" makes it sound like the AML was stationary or attached to the vehicles. I suggest something like, "the AML followed behind target diesel vehicles," instead.

We thank the reviewer for the suggestion, the change has been made.

12. p. 5, lines 1-2: Rewrite "we have referred rBC to BC in this manuscript

The change has been made from "…, we have referred rBC to BC in this manuscript" to: "…, we refer to rBC as BC in this manuscript".

13. p. 5, line 2: "detection limit" should be "detection limits".

The change has been made.

14. p. 7, line 26: Change "observed" to "reported".

The word has been changed.

---

## Author Response (AR1)

**Letter to the Editor:**

Dear Editor,

We would like to thank the two reviewers for their careful reading and constructive comments. We have revised the manuscript, incorporating the comments from the two reviewers, as noted in the two documents submitted.

Thank you again for your consideration.

Best regards,
Luisa Molina

Anonymous Referee #1

General comments

Description: This discussion paper describes emission factors of diesel-powered trucks and buses in Mexico City measured using both the Aerodyne mobile laboratory and on-road remote sensing. The targeted compounds include CO, NOx, SO2, selected VOCs, PM, black carbon (BC), and particulate organic carbon (OC). The two methods produced similar results. BC emission factors were consistent with those measured in other studies, while the OC/BC ratio was larger than found in California. Emission factors generally agreed with those used in the EPA MOVES-2014b model for NOx and BC and were higher for CO, OC, and selected VOCs.

Relevance:

Heavy duty diesel-powered vehicles are responsible for substantial amounts of BC and NOx emissions, yet there are limited on-road measurements of emissions from these vehicles. This work adds to the database of such measurements and shows that the chasing method with a mobile lab and the on-road remote sensing method produce comparable results, so it is fair to synthesize results across these different types of studies.

Assessment: The work contributes useful information about emissions from diesel engines. The writing and figures are very clear and informative. The paper illustrates the strengths and weaknesses of each of the two methods for measuring emission factors. The paper could be strengthened through some reorganization of the Results and Discussion and addition of statistical tests.

We thank the reviewer for the constructive comments on the paper.

Specific comments

1. p. 4, line 10: A little more information about the prescribed driving routes and operation of the vehicles would be useful. What was the range of speeds? Were the engines always warmed up beforehand?

We thank the reviewer for this suggestion as this will allow the results to be more adequately compared to future studies of emission characteristics of diesel vehicles in Mexico. We have now included a more detailed description of the driving conditions (the range of speeds and accelerations of the vehicle sampled) during the tests in the supplemental material document.

2. p. 7, lines 24-26: "Since the measurements were obtained in similar prescribed driving routes, the results show a wide range of average emission factors associated with each vehicle engine and emission control characteristics." The wording and logic are not quite right here. I think the authors intend to emphasize that driving conditions were very similar for all vehicles, so differences must reflect variability

between engines and control systems. But later, they assert that there is large variability even for the same vehicle.

We thank the reviewer for calling our attention to this ill-constructed phrase. As pointed out, we want to emphasize that the driving conditions were very similar for all vehicles. We have modified the paragraph accordingly:

"Since the measurements were obtained under similar prescribed driving routes, differences in results mainly reflect variability among vehicle engines and emission control characteristics."

As stated, the results indicate that even after controlling for driving routes the observed variability of emission factors still can be large. This is in agreement with current understanding of real-world emissions as compared to laboratory-based studies and the growing acknowledgment that engine performance can produce large variability under real-world operation conditions.

3. p. 8, line 7: The comparison of emission factors among different vehicle types begs for statistical tests of differences.  This is true for the presentation of differences by control technology, too.

We thank the reviewer for this valuable suggestion. We have now performed statistical testing for the significance of the results between vehicle types, by control technology, and between measurement techniques.

We have analyzed the statistical significance between control technologies for $PM_{2.5}$ EF using non-paired non-parametric Wilcoxon Rank tests and found that with a 95% confidence level the results for the EPA98 and EPA04 are significantly different. Similarly, differences between EURO3, EURO4, and EURO5 EF were found to be significantly different with a 95% confidence level. However, the rest of the tests indicated that the results for the EPA98 and the EURO3 (the older technologies sampled) were not significantly different with a 95% confidence level. Therefore, the following paragraph has been included in the text:

"Non-paired Wilcoxon Rank tests indicate that there is statistically significant difference (at the 0.05 significance level) between the $PM_{2.5}$ emission factors obtained for the EPA98 and EPA04 control technologies as well as among the EURO3, EURO4, and EURO5 technologies. However, the results for the EPA98 and the EURO3 technologies were not significantly different."

We similarly performed non-paired Wilcoxon Rank tests for comparing the emission factors by vehicle type for each pollutant. We found that CO, $NO_x$, and $SO_2$ from service trucks, urban buses, and Metrobuses were significantly different among them, whereas their VOCs measured ($C_2H_2$, acetaldehyde, benzene, toluene, C2-benzenes), and PM components (BC, OC, and inorganics) were not statistically significantly different. On the contrary, VOCs and PM-components emission factors obtained from Turibuses were statistically different from the corresponding emission factors from service trucks, urban buses, and Metrobuses. Thus we have included the following paragraph:

"Non-paired Wilcoxon Rank test indicate that there is statistically significant difference (at the 0.05 significance level) between emission factors from service trucks, urban buses, and Metrobuses for the CO, $NO_x$, and $SO_2$ pollutants, whereas their corresponding VOCs, BC, OC, and PM-inorganic emission factors were not significantly different. VOCs, BC, and PM-inorganic emission factors from biodiesel-fueled Turibuses were significantly different from the corresponding emission factors from service trucks, urban buses, and Metrobuses."

In addition to the analysis suggested by the reviewer we also evaluated the statistical significance of the results for CO and NO emission factors that were obtained with both the chasing and the remote sensing techniques. Since these represent co-sampled data we used paired t-test with a 0.05 significance level. The results indicate that in both cases of CO and NO co-samplings there is no significant difference between the results obtained by the two measurement techniques, with a confidence level of 95%. Therefore, we have now added the following paragraphs to the results:

"Paired t-tests indicate that there is no statistical significant difference (at the 0.05 significance level) between the two measurement techniques for both cases of CO and NO emission factors."

4. p. 8, line 26: The paragraph about the limitations of the sample size should be moved to the Discussion section.

As suggested by the reviewer, we have moved the discussion on the limitations of the sample size to the Discussion section.

5. p. 9, line 5: The comparison between the two methods in Section 4.1 seems it belongs more in the Results section than in the Discussion section because it is a straightforward presentation of results that address one of the objectives of the study.

We have moved the comparison of the two methods to the Results section.

6. p. 9, line 11: For comparison of the two methods, the authors chose to use the 10 seconds of AML data leading up the instant of remote sensing, which lasted 1 second. Why not isolate the 1-2 seconds of AML data that best correspond to when the remote sensing measurement was captured?

For the estimation of the emission factors using the chasing technique, the second-by-second measurements are integrated over a time period to account for the dispersion of the emission plume. Thus, if too few data points are included in the integration the resulting emission factor may not properly reflect the plume development and unnecessary uncertainty is introduced in the analysis. Based on our past experience with data analysis of this technique, we consider a good conservative

number of data points for plume development is about 10 seconds as the basis for the choice of integration time. Thus, we have now complemented the following sentence:

"Since the remote sensing technique measures the emission factor of the sampled vehicle only while it passes through the detectors, only the emission factors obtained with the mobile laboratory ~10 seconds before and up to the corresponding actual moment of co-sampling with the remote sensing detector were considered for the comparison between the two techniques. Thus, we assume that a time period of 10 seconds is sufficient to capture a large portion of the emission plume sampled by the mobile laboratory."

7. p. 11, line 15: I assume that all the vehicles tested in this study were running on petroleum diesel, so results for B10 and B20 biodiesel are irrelevant to the present study and do not merit mention here, or they require greater justification for inclusion in the comparison.

In the discussion section of the paper we compare our results on the emission factors from biodiesel vehicles to the only other available literature study of similar measurements in Mexico that used B10 and B20 blends. We believe that, given the very limited information currently available, the comparison information and discussion presented is a valuable addition and thus we have decided to keep it in the manuscript.

8. p. 12, line 10: Can the authors comment on why there are differences in the OC/BC ratio compared to that found in other studies? Might altitude explain some of it or dilution? The mobile lab and remote sensing detect fresher, less diluted plumes compared to tunnel studies.

There are several possible reasons why the results show higher OC/BC ratios in comparison to other studies. These include differences in conditions derived from the environment (e.g., altitude, temperature), technical sampling methods (capturing fresh versus more diluted emissions), and diesel fuel composition. Unfortunately it is not possible from our results to quantitatively assess the contributions from these factors as it is beyond the scope of this study. Dedicated experiments controlling for these factors as well as vehicle technology and driving conditions could help to quantify the impacts of these factors. Nevertheless, it is possible to argue that the higher OC/BC content in the Mexican results obtained with the mobile laboratory are not due to differences in the sampling technique used in tunnel studies. As the reviewer pointed out, the former capture more fresh emissions than tunnel studies and, therefore, secondary formation of organic aerosols in the air masses would only increase the OC/BC during in the tunnel study sampling (Massoli et al., 2012), which is in the opposite direction needed to explain the observed differences.

No samples were obtained in our study of the diesel fuel employed, and thus it is not possible to know its exact organic chemical composition and its effects on emissions. Although a detailed chemical composition of diesel fuel by PEMEX (Mexican National Oil Company) is not publicly available, an official

report indicates a predominant fraction of paraffin compounds of linear molecular chains with 11 to 12 carbons and a maximum 30% (in volume) of aromatics (IMP, 2014). In principle, a dedicated experiment could be set up to investigate the effects of OC formation due to differences in PEMEX's diesel fuel composition, but this is beyond the scope of this study. We have therefore included the following paragraph:

"Several factors including driving conditions, vehicle technology, and diesel fuel composition can contribute to the observed differences, but the quantification of these contributions is beyond the scope of this study. Nevertheless, the higher organic content of the emissions in the sampled Mexican vehicles with respect to those measured in California by Dallman et al., (2014) illustrate the large emission differences in PM composition that can be found in diesel fleets around the world, thus further indicating the need for locally adjusting the emission factors databases in mobile emission models."

The word has been changed.

Anonymous Referee #2

This is a very good paper with valuable information about diesel engine emission factors that can be used to prepare Mexican emission inventories. The experimental techniques used have been proven before and the results published in scientific journals. The paper is well written and the presentation of results is good. There are several questions I suggest to clarify before the paper is considered for publication. I enclose the reviewed file with comments and questions for your consideration. Please let me know if there is any doubt about my comments.

We thank the reviewer for the constructive comments on the paper.

P2.L12. You do not measure emission factors, you measure emissions and then estimate emission factors.

We agree with the reviewer that technically speaking the emissions factors were not directly measured but estimated from the measurements. Perhaps incorrectly, the literature on this topic traditionally does not make the explicit distinction between the two but it is left to the reader to grasp it from the description of the methodology. However, we do agree that it helps to the clarity of the discussions to explicitly refer to the results as estimations. We have modified the sentence accordingly from: "Compared to gaseous pollutants emissions, direct measurements of emission factors for PM components from diesel-powered vehicles are less abundant" to "Compared to gaseous pollutants emissions, measurement-based estimations of emission factors for PM components from diesel-powered vehicles are less abundant".

P3L21. Do you measure emission factors using a Mobile Laboratory? I think that you measured emissions and then you calculated emission factors with some uncertainty.

Based on the response to the previous comment, we have modified the corresponding sentence from:

"In this pilot study we measured the fuel-based emission factors for BC, OC, CO, NOx, and selected VOCs under real-world driving conditions for 20 on-road diesel vehicles in Mexico using the Aerodyne Research Inc. mobile laboratory (AML). The emission factors of NO, CO, HC, and fine PM were simultaneously measured using the cross-road remote sensing technique…"

To:

"In this pilot study we have estimated the fuel-based emission factors for BC, OC, CO, NOx, and selected VOCs under real-world driving conditions for 20 on-road diesel vehicles in Mexico using the Aerodyne

Research Inc. mobile laboratory (AML). The emission factors of NO, CO, HC, and fine PM were simultaneously obtained using the cross-road remote sensing technique…"

P4L114. Include a short description of the tests. Are these tests representative of driving conditions on the city?

We thank the reviewer for this valuable suggestion. We have now included a more detailed description of the driving conditions (the range of speeds and accelerations of the vehicle sampled) during the tests in the supplemental material document. Thanks to this suggestion, we believe that the results can be more adequately compared to future studies dedicated to understanding the emissions characteristics of diesel vehicles in Mexico.

To the best of our knowledge, there are no studies on the characteristics of driving cycles for diesel vehicles in Mexico and thus it is not possible to assess the representativeness of the tested driving conditions. In this pilot study we focused instead on sampling the selected vehicles in slow to medium speeds with frequent acceleration and deceleration periods as we anticipate these are common driving conditions in Mexico City routes. It is worth noting that the vehicles were driven by actual drivers from the participant institutions that volunteered their vehicles.

P6L27. Please explain this part being consistent with the definition of mass or moles. I understand that if the volume is the same, it is assumed that the mass of CO2 and CO is very large compared with the mass of other species, then the emission factor is the mass of species i/mass of carbon in CO2 and CO multiplied by the mass of carbon/mass of fuel, in this case (0.87), not the mass of CO2 and CO. The equation is correct if you use moles, but then, you need to include the molecular weight to obtain mass. You comment this for each technique, but the equation as presented is not consistent with the definitions.

The analysis methods used to obtain emission factors from the chasing and remote sensing techniques are well established and reported in detail in the references provided in the paper. As pointed out by the reviewer, the molecular weights are needed to convert from the measured gaseous species (in moles) to mass. For the AML technique we believe this is already explained in the following sentence:

"For the AML technique the gaseous species mass ratio is obtained using the moles of the pollutant and the total moles of emitted carbon by multiplying with their respective molecular weights, whereas the PM components measurements are directly obtained in $\mu g m^{-3}$, therefore the denominator units for the total carbon content need to be converted accordingly to $\mu g C m^{-3}$."

To add clarity, we have completed the last phrase as follow: "…therefore the denominator units for the total carbon content need to be converted accordingly to $\mu g C m^{-3}$ using the respective $CO_2$ and CO molecular weights." In this way, it is clearer that for the AML technique the equation is used only after the measured (in moles) carbon content is transformed to mass.

For the application of equation 1 in the remote sensing technique, the reviewer correctly pointed out that more clarity is needed. We have expanded the sentence: "For the RS technique, the ratio is obtained from the differences in number of molecules measured before and after the sampled vehicle" to: "The RS technique measures the difference in the number of molecules of the pollutant $i$ before and after the sampled vehicle passes through the detector, thus the fuel-based emission factor is estimated from the ratio of emitted $n$ moles of $i$ to $n$ moles of $CO_2$ ($n_i/n_{CO2}$), dividing it by the sum of carbon moles in the $CO_2$, CO, and HC ratios to $CO_2$ (1, $n_{CO}/n_{CO2}$, and $n_{HC}/n_{CO2}$, respectively), and multiplying by the corresponding molecular weights and $w_c$."

P6L29. What is the source of information of carbon content in diesel? Is the value in mass fraction or in mole fraction?

The value of $w_c$ is in mass fraction. The selection of 0.87 as the carbon content in diesel fuel is a common assumption based on typical compositions that consider it as 11 to 22 carbon linear chains. PEMEX does not provide a generalized chemical formula for diesel but we consider that a good H to C ratio assumption is 1.8.

We have now clarified this by adding the words "assumed as" when referring to this value and replacing the words "weight fraction" by "mass fraction" in the sentence: "… normalized by the mass fraction carbon content of the diesel fuel $w_c$ (assumed as 0.87) as shown in Eq. (1)"

P9.L15. In the figure it looks like the calculated values of CO EF are closer to the 1:1 ratio than the NO EF values, and I would expect a better agreement between the two techniques, but you argue the opposite. Please explain. The scale for CO data is larger than for NO data, then it is expected that the variance for CO is higher. The R2 is a comparison with respect to the mean, but I think you should be comparing with respect to a straight line with slope 1. Could you elaborate on this please?

In Figure 4 we compare individual CO and NO emission factors whenever they were obtained simultaneously by the two techniques. In the discussions of this comparison, our point is that although both techniques essentially present similar co-variability (that is, both techniques capture low and high emissions conditions) there are important differences between the results by vehicle type. The reviewer is right in that it is better to focus on the comparison of the linearity of the data. To add clarity in the text, we have replaced the discussions on the coefficient of determination for the values of the Pearson linear correlation coefficient and have added the comparison of the slope of the data to a 1:1 ratio.

As the reviewer pointed out, we argue that NO values show a linear but disperse correlation whereas CO values show less of a linear correlation (the overall Pierson coefficient for linear correlation for NO is 0.75 and slope 0.77:1 whereas the linear correlation coefficient is 0.60 for CO with a slope 0.25:1; although, as discussed in the text, both linear correlations can vary substantially by vehicle type). This is because the CO data presented in the figure is in logarithmic scale for clarity as the values can vary up to 2 orders of magnitude. Thus, we have replaced the discussion paragraph as follows:

"Whereas the overall Pearson linear correlation coefficient (R) between the two techniques is only 0.60 (slope 0.25:1) for CO, the coefficient increases to 0.96 and 0.92 for Metrobuses and service trucks, respectively. The lower CO emission factors for the UB1 high-emitter measured by the RS in comparison with the AML contributed significantly to the overall small linear correlation coefficient:  R increases to 0.78 if the CO data for the UB1 is not included in the comparison. Similarly, the overall R for NO emission factors between the two techniques is only 0.75 (slope 0.77:1) but it increases to 0.86 and 0.84 for service trucks and urban buses, respectively."

P10L29. Why turibuses are not included? Does MOVES have emission factors for turibuses?

Turibuses are not included as an explicitly vehicle category in MOVES and thus we have not used them in the comparisons.

P11L31. Describe DPF. I did not find it in the paper.

We thank the reviewer for calling our attention to this missing definition. We have now added the definitions of DPF (diesel particle filter) and selective catalytic reduction (SCR) in the text and in Table 3 where they are used.

[revised manuscript text omitted]

**2. Sampling driving conditions**

To the best of our knowledge, there are no studies on the characteristics of driving cycles for diesel vehicles in Mexico and thus it is not possible to assess the representativeness of the tested driving conditions. In this pilot study we focused instead on sampling the selected vehicles in slow to medium speeds with frequent acceleration and deceleration periods as we anticipate these are common driving conditions in Mexico City routes.

The selected vehicles were sampled using similar driving conditions by following the same route and driving under similar ranges of speeds and accelerations multiple times. The vehicles engines were previously warmed up before each measurement, and thus the measurements do not represent cold-start emissions conditions. In addition, all RTP buses, Metrobuses and Turibuses vehicles were sampled in full load capacity with the kind collaboration of volunteer students (an exception was the single DINA bus sampled, which was ballasted using filled water cans), whereas service trucks were ballasted with actual goods provided by the participating institutions. The sampled vehicles were driven by actual drivers from the corresponding participant institutions.

As described in the main manuscript, a global positioning system (GPS) was used on-board the mobile laboratory to obtain the spatial coordinates during the study. Since the measurements are obtained in vehicle "chase" mode, at a first approximation these data can be used to describe the speed and acceleration driving conditions. The average time for a given driving cycle was of 3.3 minutes with an average speed of 5.5 m/s. To assess the fraction of the time that the measurements are obtained in acceleration, deceleration, or cruising modes it is necessary to define a speed change criteria over the GPS data acquisition time (1 second). Following the

procedure of Tong et al (2000) we have defined the acceleration, deceleration, and cruising modes as follow:

1) Acceleration mode: positive incremental speed changes of more than 0.1 m/sec/sec during the 1-second interval.
2) Deceleration mode: negative incremental speed changes of more than to 0.1 m/sec/sec during the 1-second interval.
3) Cruising mode: absolute incremental speed changes of less than or equal to 0.1 m/sec/sec during the 1-sec interval.

The resulting driving cycle distribution is shown in Table S3.

Table S3. Summary characteristics of sampling driving cycles.

| Mode | % of time |
|------|-----------|
| Acceleration | 22.3 |
| Deceleration | 34.8 |
| Cruising | 29.1 |
| Idling | 13.8 |

The following figures are discussed in the text of the manuscript. As a note, the emissions of CO, $NO_x$, HC and PM for a Dina vehicle were further co-sampled using an AXION PEMS instrument (see Figure S1). Therefore, for this vehicle the chasing, cross-road remote sensing, and PEMS techniques were applied simultaneously. However, the results of the inter-comparison of the three techniques are not included in this manuscript but are discussed in separate publication.

[Figure]

Figure S1 Top figure shows an aerial photo of the Modulo 23 of the RTP facilities with an indication of the location of the remote sensing unit and the area for the chasing experiments. Photos on the right show a service truck passing through the remote sensing detectors unit (top-right photo) and the Dina bus sampled with the mobile laboratory, remote sensing and PEMS techniques.

[Figure]

Figure S2. Examples of the four vehicle types (Metrobus, Turibus, urban RTP bus, and service truck) sampled in this pilot study.

[Figure]

Figure S3. Top panel shows the number of diesel-powered vehicles by model year for the Mexican fleet for the year 2013. Trucks are classified by gross vehicle weight (GVW). Bottom panel shows the corresponding percentage of the number of diesel powered vehicles by model year. Source: prepared from data from the 2013 Mexican Nacional Emissions Inventory (SEMARNAT, 2015).

[Figure]

Figure S4. Top and bottom panels show the time evolution of number of gasoline-powered vehicles and diesel-powered vehicles, respectively, by model year (MY) in Mexico City. The figure shows a more rapid decline in the number of older gasoline vehicles than of diesel vehicles. Thus, older diesel vehicles remain in-use in the fleet for much longer periods than the gasoline vehicles. Source: prepared from data from the 2014 Mexico City Metropolitan Area Emissions Inventory (SEDEMA, 2017).